# Ankyrin-R regulates fast-spiking interneuron excitability through perineuronal nets and Kv3.1b K$^+$ channels

**Sharon R Stevens[1], Colleen M Longley[2,3], Yuki Ogawa[1], Lindsay H Teliska[1], Anithachristy S Arumanayagam[4], Supna Nair[5], Juan A Oses-Prieto[5], Alma L Burlingame[5], Matthew D Cykowski[4], Mingshan Xue[1,2,3,6], Matthew N Rasband[1,2]***

[1]Department of Neuroscience, Baylor College of Medicine, Houston, United States; [2]Program in Developmental Biology, Baylor College of Medicine, Houston, United States; [3]The Cain Foundation Laboratories, Jan and Dan Duncan Neurological Research Institute at Texas Children's Hospital, Houston, United States; [4]Department of Pathology and Genomic Medicine, Houston Methodist Hospital, Houston, United States; [5]Department of Pharmaceutical Chemistry, University of California San Francisco, San Francisco, United States; [6]Department of Molecular and Human Genetics, Baylor College of Medicine, Houston, United States

*For correspondence:
rasband@bcm.edu

Competing interests: The authors declare that no competing interests exist.

**Abstract** Neuronal ankyrins cluster and link membrane proteins to the actin and spectrin-based cytoskeleton. Among the three vertebrate ankyrins, little is known about neuronal Ankyrin-R (AnkR). We report AnkR is highly enriched in Pv$^+$ fast-spiking interneurons in mouse and human. We identify AnkR-associated protein complexes including cytoskeletal proteins, cell adhesion molecules (CAMs), and perineuronal nets (PNNs). We show that loss of AnkR from forebrain interneurons reduces and disrupts PNNs, decreases anxiety-like behaviors, and changes the intrinsic excitability and firing properties of Pv$^+$ fast-spiking interneurons. These changes are accompanied by a dramatic reduction in Kv3.1b K$^+$ channels. We identify a novel AnkR-binding motif in Kv3.1b, and show that AnkR is both necessary and sufficient for Kv3.1b membrane localization in interneurons and at nodes of Ranvier. Thus, AnkR regulates Pv$^+$ fast-spiking interneuron function by organizing ion channels, CAMs, and PNNs, and linking these to the underlying β1 spectrin-based cytoskeleton.

## Introduction

Ion channels and cell adhesion molecules (CAMs) are frequently recruited to, stabilized, and maintained at specific neuronal membrane domains by scaffolding proteins. The ankyrin scaffolding proteins, consisting of Ankyrin-R, -B, and -G (AnkR, AnkB, and AnkG, respectively), are the primary link between the submembranous spectrin-based cytoskeleton and the cytoplasmic domains of many transmembrane proteins (*Michaely and Bennett, 1995*; *Sedgwick and Smerdon, 1999*). Ankyrins share a common structural organization including an N-terminal membrane binding domain consisting of 24 ankyrin repeats, a spectrin-binding domain also known as the ZZU domain, a death domain, and a more divergent and unstructured C-terminal domain that modulates interactions with other cytoskeletal proteins. Alternative splice variants of AnkB and AnkG arise from giant exon insertion between the spectrin binding domain and the death domain (*Bennett and Lorenzo, 2013*; *Stevens and Rasband, 2021*). Much is known about the functions of AnkG and AnkB in the nervous system. For example, AnkG links Na$^+$ and K$^+$ channels, and the CAM neurofascin 186 at axon initial segments (AIS) and nodes of Ranvier to the underlying β4 spectrin and actin-based cytoskeleton.

The clustering of channels at the AIS and nodes facilitates fast and efficient action potential propagation (*Dzhashiashvili et al., 2007*; *Zhou et al., 1998*). Similarly, AnkB stabilizes Na$^+$ channels and L1CAM family membrane proteins in unmyelinated axons and at paranodal junctions of myelinated axons by linking these membrane proteins to β2 spectrin (*Chang et al., 2014*; *Scotland et al., 1998*; *Susuki et al., 2018*). However, little is known about the function of AnkR in the nervous system. Instead, AnkR has mostly been studied in red blood cells where it maintains the cell's structural integrity via its link between β1 spectrin and the cytoplasmic domain of the anion exchanger Band 3 (*Bennett and Stenbuck, 1979*). Loss of AnkR results in fragile erythrocyte membranes and hemolytic anemia (*Lux et al., 1990*). Intriguingly, case studies of patients with hereditary spherocytic anemia, caused by mutations in AnkR, report various neurological disturbances (*Coetzer et al., 1988*; *McCann and Jacob, 1976*; *Miya et al., 2012*), and a number of recent epigenome-wide association studies in Alzheimer's disease have consistently found neuropathology-associated DNA hypermethylation of *ANK1* (*ANK1* is the gene encoding AnkR) (*De Jager et al., 2014*; *Gasparoni et al., 2018*; *Higham et al., 2019*; *Lunnon et al., 2014*; *Smith et al., 2019a*; *Smith et al., 2019b*). Additionally, AnkR can substitute for AnkG to cluster Na$^+$ channels at nodes of Ranvier (*Ho et al., 2014*) but not AIS (*Liu et al., 2020*). Thus, AnkR may play important, but as yet undefined, roles in nervous system function in both the healthy and diseased brain.

In addition to clustering ion channels, and through its interaction with CAMs, AnkG assembles and maintains a complex extracellular matrix (ECM) consisting of chondroitin sulfate proteoglycans and other ECM proteins that surround AIS and nodes of Ranvier (*Amor et al., 2017*; *Hedstrom et al., 2007*; *Susuki et al., 2013*). Thus, ankyrins may function generally to link ECMs to the cytoskeleton through their membrane receptors. One highly condensed and specialized ECM in the nervous system is the perineuronal net (PNN). PNNs surround synaptic innervations and are thought to be important to maintain the balance of excitation and inhibition (*Carceller et al., 2020*). The majority of PNNs surround the soma and proximal dendrites of fast-spiking parvalbumin-positive (Pv$^+$) inhibitory interneurons and have a chondroitin sulfate proteoglycan composition similar to the perinodal and axon initial segment extracellular matrix (*Fawcett et al., 2019*). However, how PNNs themselves are assembled, maintained, and restricted to specific domains and neuronal subtypes is unknown.

Here, we show the loss of AnkR from GABAergic forebrain neurons results in a reduction and altered structure of PNNs, a reduction in anxiety-like behaviors, and altered intrinsic excitability and firing properties of PNN$^+$ fast-spiking interneurons. We identify AnkR-interacting adhesion molecules that may tether PNNs to the spectrin cytoskeleton. Importantly, the altered excitability reflects the loss of Kv3.1b K$^+$ channels. We identify the motif in Kv3.1b necessary for its interaction with AnkR. We show AnkR is both necessary and sufficient for the recruitment and clustering of Kv3.1b K$^+$ channels in the neuronal membrane.

## Results

### AnkR is highly enriched in Pv$^+$ inhibitory interneurons

To determine where AnkR is located, we immunostained mouse forebrain using antibodies against AnkR. We found that AnkR is highly enriched in the perisomatic region of a subset of neurons sparsely distributed throughout the cortex and hippocampus (*Figure 1A*). Immunoblotting of brain homogenates shows AnkR protein in the brain increases during early postnatal development, reaching mature expression levels by postnatal day 30 (*Figure 1B*). The sparse distribution of AnkR-labeled cells was highly reminiscent of the distribution of cortical interneurons. Indeed, immunostaining with antibodies against parvalbumin (Pv), a marker of fast-spiking interneurons (*Figure 1C,D*, and *Figure 1—figure supplement 1A*), shows >90% of Pv$^+$ cells in adult cortex and hippocampus have high levels of AnkR, while only 79% of AnkR$^+$ cells in cortex and 64% in hippocampus are Pv$^+$. Somatostatin (SST) expressing interneurons account for some of the remaining AnkR$^+$ cells, with SST expressed in 8% of AnkR$^+$ neurons in cortex and 22% of AnkR$^+$ cells in hippocampus (*Figure 1C, D*). AnkR is also highly expressed in Pv$^+$ neurons in human brain. AnkR staining was identified in laminae II-VI in human cortical biopsy samples. Labeling was strong and membranous and was most intense in larger neurons of laminae III and V (*Figure 1E*). AnkR staining intensity in large neurons of lamina III and V was similar to staining intensity of erythrocytes (not shown). Furthermore, AnkR labeling

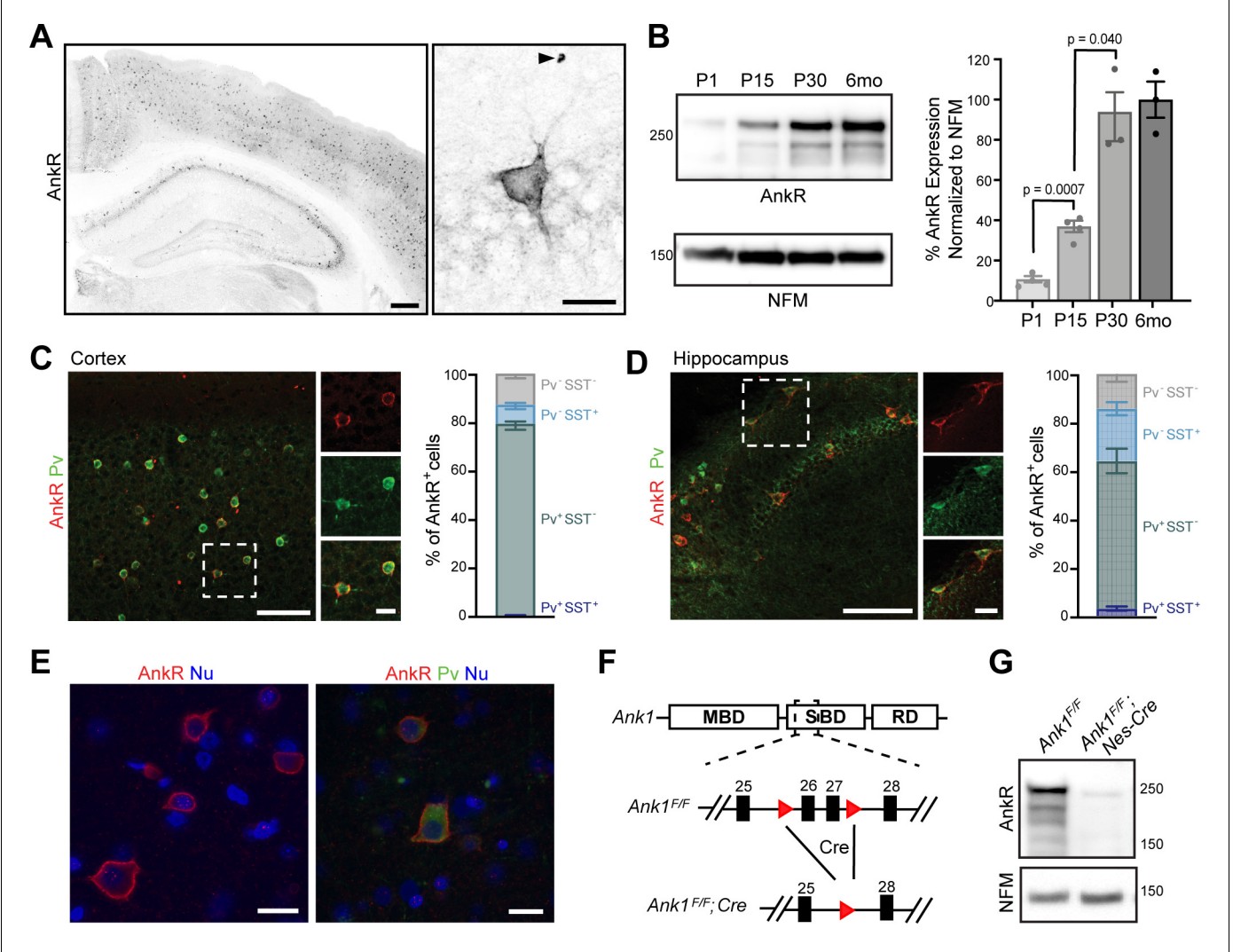

**Figure 1.** AnkR is expressed in select neurons in the cortex and hippocampus. (**A**) Immunostaining of coronal mouse brain for AnkR. Arrowhead indicates a red blood cell. Scalebars, left, 250 μm, right, 20 μm. (**B**) Immunoblot of P1, P15, P30, and 6-month C57BL/6 brain homogenates for anti-AnkR and neurofilament-M (NFM). Quantification of immunoblot in ImageJ by comparison of AnkR to NFM loading control from three independent duplicate experiments; P1, P15 (N=four mice), P30, 6 months (N=three mice). Samples were normalized to NFM, then compared to 6-month animals. Statistical significance was determined by multiple unpaired t-tests with Welch correction and Holm-Šídák method for multiple comparisons. (**C, D**) Immunostaining for AnkR (red) and Parvalbumin (Pv, green) in 3-month cortex (**C**, N = three mice; n=923 cells) and hippocampus (**D**, N = three mice; n=342 cells). Scalebars, 50 μm; inset, 10 μm. (**E**) Immunostaining of human cortex for AnkR (red) and Parvalbumin (green). Nuclei are labeled using DAPI (blue). Scalebars, 20 μm. (**F**) Schematic of the *Ank1* conditional allele. *loxP* sites (red triangles) flank exons 26 and 27 in the spectrin-binding domain (SBD), after the membrane-binding domain (MBD) and before the regulatory domain (RD). Cre-mediated recombination and removal of exons 26 and 27 generates a premature stop codon in exon 28. (**G**) Immunoblot of brain homogenates for anti-AnkR, and Neurofilament-M. Molecular weights are indicated at right in kDa. All error bars indicate mean ± SEM.

The online version of this article includes the following source data and figure supplement(s) for figure 1:

**Source data 1.** Source data related to *Figure 1* .

**Figure supplement 1.** AnkR immunostaining in control and *Ank1* conditional knockout mice.

was only identified in a subset of neurons in human cortex and staining was not present in endothelial cells or glia. A subset of AnkR⁺ neurons in laminae III and V co-expressed Pv in the neuronal cytoplasm (*Figure 1E*), although, as in mice, the total number of AnkR⁺ positive neurons exceed the number of Pv⁺ cells. These results expand on those previously reported in rat neocortex (*Wintergerst et al., 1996*).

To determine the role of neuronal AnkR and confirm its high expression in interneurons, we constructed a floxed allele for *Ank1* (*Ank1^F/F^*; *Figure 1F*); this new model allows for an exploration of AnkR function in the brain while avoiding the confound of anemia due to loss of AnkR from red blood cells. We removed AnkR from the nervous system using *Nestin-Cre* mice (*Ank1^F/F^;Nes-Cre*), and from GABAergic forebrain interneurons using *Dlx5/6-Cre* mice (*Ank1^F/F^;Dlx5/6-Cre*). We confirmed the efficient loss of AnkR after recombination by immunoblot of *Ank1^F/F^;Nes-Cre* brain homogenates (*Figure 1G*). Moreover, immunostaining of forebrain sections from both *Ank1^F/F^;Nes-Cre* and *Ank1^F/F^;Dlx5/6-Cre* showed efficient and selective loss of AnkR in neurons; AnkR expression in erythrocytes was unaffected (*Figure 1—figure supplement 1B*). These results also highlight the specificity of our antibodies. Additionally, hemoglobin levels in *Ank1^F/F^;Nes-Cre* and *Ank1^F/F^;Dlx5/6-Cre* mice were normal (data not shown). Although most Pv$^+$ cells are AnkR$^+$ (*Figure 1C,D*), there is a subpopulation of AnkR$^+$ neurons that are not Pv$^+$ or SST$^+$; these cells are most likely another subtype of GABAergic neuron since neurons with high levels of AnkR immunoreactivity in cortical and hippocampal neurons were not detected in *Ank1^F/F^;Dlx5/6-Cre* mice (*Figure 1—figure supplement 1B*). Interestingly, and in contrast to the rescue of AnkG by AnkR at nodes of Ranvier (*Ho et al., 2014*), we found no evidence for reciprocal compensation by AnkG in AnkR-deficient neurons. AnkG in Pv$^+$ neurons in cortex remained highly restricted to the AIS and nodes in all genotypes analyzed (*Figure 1—figure supplement 1C* and data not shown). Together, these results show that AnkR is abundantly expressed in Pv$^+$ interneurons of the forebrain, that its localization is distinct from that of AnkB and AnkG, and that we have generated a floxed *Ank1* allele that allows for cell-type-specific deletion in the nervous system.

## Elucidating the AnkR interactome

What functions does AnkR have in GABAergic forebrain interneurons? To begin to answer this question we determined AnkR's interactome. Since AnkR and β1 spectrin are binding partners in erythrocytes and can function together at nodes of Ranvier to stabilize Na$^+$ channels (*Liu et al., 2020*), we first showed that AnkR and β1 spectrin also form a protein complex in Pv$^+$ interneurons. Immunostaining showed that β1 spectrin is highly expressed and colocalizes with AnkR in forebrain interneurons (*Figure 2A*). Furthermore, β1 spectrin and AnkR reciprocally co-immunoprecipitate each other (*Figure 2B*). Immunoblots of *Ank1^F/F^;Nes-Cre* mouse brain homogenates show that β1 spectrin protein levels are significantly reduced in the absence of AnkR compared to control mice (*Figure 2C*). Similarly, immunostaining of cortex and hippocampus from *Ank1^F/F^;Nes-Cre* mice showed remarkably reduced β1 spectrin immunofluorescence compared to control mice (*Figure 2—figure supplement 1A,B*). Together, these results show that AnkR interacts with and maintains β1 spectrin in forebrain interneurons.

To identify additional AnkR interacting proteins, we combined two unbiased mass spectrometry screens (*Figure 2—figure supplement 1C*). First, we performed mass spectrometry on AnkR immunoprecipitations (IPs) in biological triplicate from whole *Ank1^F/F^* mouse brain lysates. These yielded 3241 unique proteins with at least one peptide spectral match (PSM) in each sample. We further narrowed the number of potential AnkR-binding proteins by setting an arbitrary threshold of ≥10 mean PSMs found in the IPs. As a second, orthogonal approach, we performed differential proteomics and mass spectrometry using wildtype (*Ank1^+/+^* or WT) and AnkR-deficient (*Ank1^pale/pale^* or AnkR-KO) (*Ho et al., 2014*) hindbrain homogenates (*Figure 2—figure supplement 1C*). We used AnkR-KO mice rather than *Ank1^F/F^;Nes-Cre* mice to avoid confounds due to incomplete recombination or contributions from cells still expressing AnkR. Mass spectrometry yielded 2465 unique proteins, of which 986 were reduced in AnkR-KO mice compared to WT. Since ankyrins function as scaffolds that stabilize and retain membrane proteins, we reasoned that loss of AnkR might result in increased turnover and lower amounts of AnkR-interacting proteins. Therefore, we focused only on those proteins that had fewer PSMs in the AnkR-KO compared to WT. To further refine our analysis, we set an arbitrary threshold such that potential AnkR-interacting proteins must have ≥20% reduction in PSMs, with ≥5 PSMs found in WT mice and a ≥3 PSMs difference between WT and AnkR-KO. Combining the two data sets revealed 72 potential AnkR-interacting proteins (*Figure 2—figure supplement 1C* and *Figure 2—source data 1*).

We sorted these 72 proteins into functional categories and plotted them using concentric rings to indicate the percent reduction in AnkR-KO mice, with circle size representing the mean number of PSMs in the IPs (*Figure 2D*). Among the proteins enriched and passing our stringent filtering criteria,

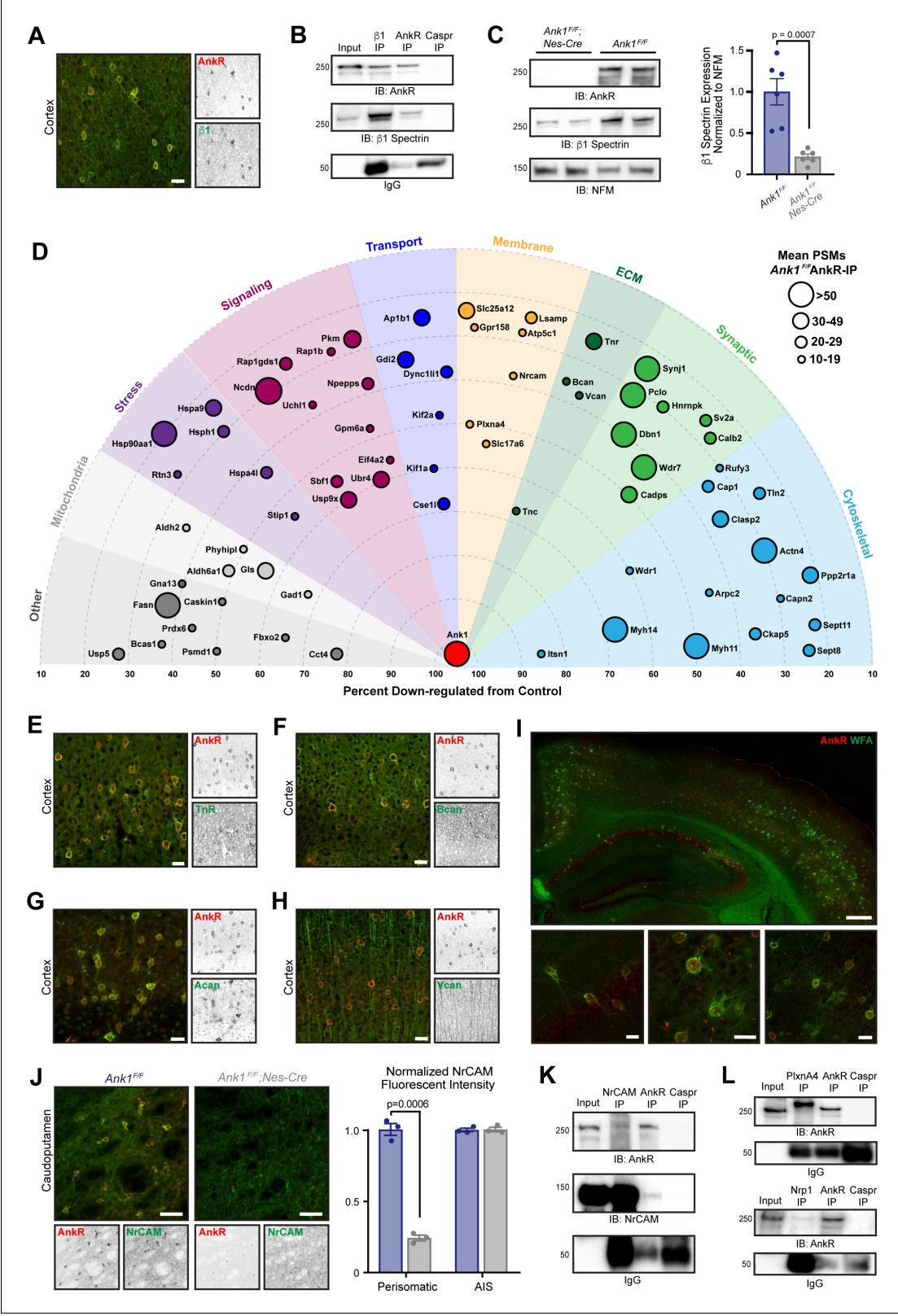

**Figure 2.** AnkR interacting proteins. (**A**) Immunostaining of mouse cortex using antibodies against AnkR (red) and β1 spectrin (green). Scalebar, 20 μm. (**B**) Immunoblot of β1 spectrin, AnkR, and IgG immunoprecipitation reactions using antibodies against AnkR and β1 spectrin. IP, immunoprecipitation; IB, immunoblot. (**C**) Immunoblot of control and AnkR-deficient mouse brains using antibodies against β1 spectrin, AnkR, and neurofilament-M (NFM). Quantification of the β1 spectrin immunoblots normalized to NFM. Error bars indicate mean ± SEM. N=6. (**D**) Top AnkR-interacting candidates. Circle

*Figure 2 continued on next page*

*Figure 2 continued*

size corresponds to the mean PSM from IP mass spectrometry. Concentric circles radiating from *Ank1* correspond to % reduction in PSMs from AnkR knockout mouse compared to control. Identified proteins are organized according to their putative functions. (**E–H**) Immunostaining of mouse cortex using antibodies against AnkR (red) and TnR (green, **E**), Bcan (green, **F**), Acan (green, **G**), and Vcan (green, **H**). Scalebars, 20 µm. (**I**) Immunostaining of mouse cortex and hippocampus using antibodies against AnkR (red) and fluorescent-labeled WFA (green). Scalebars, 250 µm (top) and 20 µm (bottom). (**J**) Immunostaining of control (left) and AnkR-deficient mouse brain (right) using antibodies against AnkR (red) and NrCAM (green). Scalebars, 50 µm. Normalized fluorescence intensity of perisomatic and axon initial segment (AIS) NrCAM in *Ank1$^{F/F}$* (blue) and *Ank1$^{F/F}$;Nes-Cre* (gray). Error bars indicate mean ± SEM. N=3. (**K**) Immunoblot of NrCAM, AnkR, and IgG immunoprecipitation reactions using antibodies against AnkR (top) and NrCAM (bottom). (**L**) Immunoblot of PlexinA4, AnkR, and IgG immunoprecipitation reactions using antibodies against AnkR (top). Immunoblot of Nrp1, AnkR, and IgG immunoprecipitation reactions using antibodies against AnkR (bottom).

The online version of this article includes the following source data and figure supplement(s) for figure 2:

**Source data 1.** Source data realted to *Figure 2* .

**Figure supplement 1.** AnkR interacting proteins.

we found cytoskeletal, membrane, signaling, and ECM proteins. Surprisingly, although AnkR is a cytoplasmic scaffolding protein, the ECM proteins TenascinC (TnC), TenascinR (TnR), Brevican (Bcan) and Versican (Vcan) were all identified in the AnkR IPs and also enriched in WT compared to AnkR-KO. We previously reported that AnkG, through the CAM neurofascin 186, interacts with and recruits the chondroitin sulfate proteoglycans Bcan and Vcan to AIS and nodes of Ranvier (*Hedstrom et al., 2008*; *Susuki et al., 2013*). TnR is also found at nodes and binds to Vcan and Bcan (*Bekku et al., 2009*). Immunostaining of cortex revealed that antibodies against TnR and Bcan strongly label AnkR$^+$ neurons (*Figure 2E,F*); Vcan immunoreactivity was not restricted to AnkR$^+$ neurons and was more widely distributed (*Figure 2H*). TnR, Bcan, and Vcan are well-known components of PNNs (*Fawcett et al., 2019*; *Wintergerst et al., 1996*). Immunostaining for the chondroitin sulfate proteoglycan and PNN protein Aggrecan (Acan) (*Carulli et al., 2007*; *Rowlands et al., 2018*) also showed strong colocalization with AnkR (*Figure 2G*); although Acan was detected by mass spectrometry, it did not pass our stringent filtering. In addition to antibodies against Bcan, TnR, and Acan, PNNs can also be detected using the fluorescently-labeled *Wisteria Floribunda* (WFA) lectin (*Brückner et al., 1993*), which binds to *N*-acetyl-D-glucosamine at the ends of chondroitin sulfate chains. Co-staining of WFA and AnkR shows that AnkR$^+$ neurons are surrounded by WFA-labeled PNNs (*Figure 2I*).

How can AnkR, an intracellular scaffolding protein, interact with extracellular PNNs? We reasoned this could occur through CAMs that bridge AnkR and PNNs. Our list of potential AnkR-binding CAMs included two strong candidates: NrCAM and PlexinA4. NrCAM is a member of the L1 family of CAMs with known ankyrin-binding activity (*Davis and Bennett, 1994*). However, NrCAM can also be shed from the cell surface and incorporated into the ECM surrounding nodes of Ranvier through direct binding to neurofascin 186 (*Susuki et al., 2013*). Thus, NrCAM can function both as a membrane receptor and as a component of the perinodal ECM. Immunostaining for NrCAM showed strong colocalization with both AnkR$^+$ and WFA$^+$ neurons in caudoputamen (*Figure 2J* and *Figure 2—figure supplement 1G*), but less robust colocalization in hippocampus and cortex (*Figure 2—figure supplement 1D–F*). These results suggest AnkR may function together with NrCAM in a subset of GABAergic neurons and emphasizes the diversity of PNNs and their interacting proteins. Perisomatic, but not AIS, NrCAM immunoreactivity was dramatically reduced in *Ank1$^{F/F}$;Nes-Cre* mouse brain (*Figure 2J*). Furthermore, NrCAM and AnkR reciprocally co-immunoprecipitate each other (*Figure 2K*). These results support recent work exploring the molecular heterogeneity of PNNs and suggests another potential PNN subtype involving a unique CAM membrane receptor (*Irvine and Kwok, 2018*; *Yamada and Jinno, 2017*).

PlexinA4 functions together with Neuropilin-1 (Nrp1) as a receptor for semaphorin signaling (*Nakamura et al., 2000*). Sema3A is a component of the PNNs surrounding Pv$^+$ interneurons (*Kwok et al., 2011*); moreover, enzymatic or genetic disruption of PNNs reduces Sema3A (*de Winter et al., 2016*). PlexinA4 and Nrp1 are widely expressed throughout the nervous system. Although immunostaining did not reveal any specific enrichment for PlexinA4 in AnkR$^+$ interneurons, immunostaining for Nrp1 showed strong enrichment in AnkR$^+$ neurons in deep cerebellar nuclei (*Figure 2—figure supplement 1I*). Nevertheless, we found that PlexinA4 and Nrp1 co-immunoprecipitated with AnkR from brain homogenates (*Figure 2L*). Together, our proteomic studies show that

AnkR co-localizes with multiple PNN proteins and may indirectly interact with PNNs through the membrane receptors NrCAM and PlexinA4. Other membrane proteins identified in our proteomics, but not analyzed here, may also function to link PNNs to AnkR. Based on these proteomic, biochemical, and immunostaining results, we focused on the relationship between AnkR and PNNs.

## AnkR is required to maintain PNN density and structure

To determine if AnkR contributes to the formation, maintenance, and structure of PNNs, we used WFA to label PNNs in cortex and hippocampus of 1-month-old $Ank1^{F/F}$, $Ank1^{+/+}$;$Dlx5/6$-$Cre$, and $Ank1^{F/F}$;$Dlx5/6$-$Cre$ mice (*Figure 3A* and *Figure 3—figure supplement 1*). At this age, we found little difference in the number of WFA$^+$/Pv$^+$, WFA$^-$/Pv$^+$, or WFA$^+$/Pv$^-$ neurons per unit area (UA) in either hippocampus or cortex (*Figure 3B*). However, when we measured the fluorescence intensity of WFA, we found a significant reduction in both cortex and hippocampus in $Ank1^{F/F}$;$Dlx5/6$-$Cre$ mice compared to floxed or Cre controls (*Figure 3C*) further subdividing cortical regions showed a similar decrease in several regions (*Figure 3—figure supplement 2A,C*). Examination of PNNs at high magnification showed that in the absence of AnkR, PNNs were less compact and disrupted compared to control mice. We classified the PNNs as being dense (0), having a few small holes (1), or having large numerous holes (2) (*Figure 3—figure supplement 2D*). AnkR-deficient neurons were significantly less likely to be compact, characterized as being more likely to have large holes in their nets, and increased WFA thickness (*Figure 3D–F* and *Figure 3—figure supplement 2E*). In 12-month-old mice, we found a significant reduction in the number of WFA$^+$/Pv$^+$ cells per unit area in the hippocampus and cortex of $Ank1^{F/F}$;$Dlx5/6$-$Cre$ mice compared to floxed controls (*Figure 3G, H*). In addition, and like in the one-month-old $Ank1^{F/F}$;$Dlx5/6$-$Cre$ mice, we measured a ~50% decrease in the WFA fluorescence intensity in both hippocampus and cortex (*Figure 3I*). This significant reduction in WFA was seen in multiple cortical regions (*Figure 3—figure supplement 2B*). The normally compact PNN structure, as observed in 12-month-old control mice, was also significantly disrupted in $Ank1^{F/F}$;$Dlx5/6$-$Cre$ mice with even more prominent holes in the PNNs, and increased WFA thickness compared to either control or one-month-old $Ank1^{F/F}$;$Dlx5/6$-$Cre$ mice (*Figure 3J–L*, *Figure 3—figure supplement 2F*, and *Videos 1* and *2*). Furthermore, the strong reduction in NrCAM immunoreactivity seen in $Ank1^{F/F}$;$Nes$-$Cre$ mouse brain (*Figure 2J*) was matched by a strong reduction in WFA$^+$ cells in the same region (*Figure 2—figure supplement 1G,H*). Together, these results suggest that loss of AnkR does not disrupt the ability of PNNs to form, but rather AnkR helps maintain PNNs and their normal compact structure through binding to PNN-interacting CAMs like NrCAM and PlexinA4.

## Loss of AnkR from GABAergic forebrain interneurons decreases anxiety-like behaviors

To determine if loss of AnkR from GABAergic interneurons alters nervous system function, we analyzed the behavior of $Ank1^{F/F}$;$Dlx5/6$-$Cre$ mice compared to control mice ($Ank1^{F/F}$ and $Ank1^{+/+}$;$Dlx5/6$-$Cre$). We first performed a 30-min open-field assessment to rule out locomotor deficits (*Figure 4A* and *Figure 4—figure supplement 1A–C*) since a hypomorph of AnkR was previously reported to have a loss of Purkinje neurons by 6 months of age (*Peters et al., 1991*). We found that all genotypes analyzed had normal locomotion, but both $Ank1^{+/+}$;$Dlx5/6$-$Cre$ and $Ank1^{F/F}$;$Dlx5/6$-$Cre$ mice showed increased velocity and distance traveled compared to $Ank1^{F/F}$ mice. These results are consistent with previous reports that the $Dlx5/6$-$Cre$ transgene results in a hyperactive phenotype characterized by increased velocity and movement (*de Lombares et al., 2019*). Nevertheless, during the open-field assessment, $Ank1^{F/F}$;$Dlx5/6$-$Cre$ mice spent significantly more time in the center of the arena and less time in the perimeter compared to both $Ank1^{F/F}$ and $Ank1^{+/+}$;$Dlx5/6$-$Cre$ mice (*Figure 4B,C*), suggesting that loss of AnkR may be anxiolytic.

To further distinguish between increased activity and decreased anxiety-like behaviors, we used the elevated plus maze (*Figure 4D–G*). In contrast to the open-field assessment, during a 10-min trial in the elevated plus maze we observed no difference among the three genotypes in the velocity or total distance traveled in all arms (*Figure 4—figure supplement 1D,E*). However, we measured a significant increase in the distance traveled in the open arms and the time spent in the open arms by $Ank1^{F/F}$;$Dlx5/6$-$Cre$ mice compared to control mice (*Figure 4E,F*). Together, these data suggest that loss of AnkR from forebrain GABAergic neurons reduces anxiety-like behaviors.

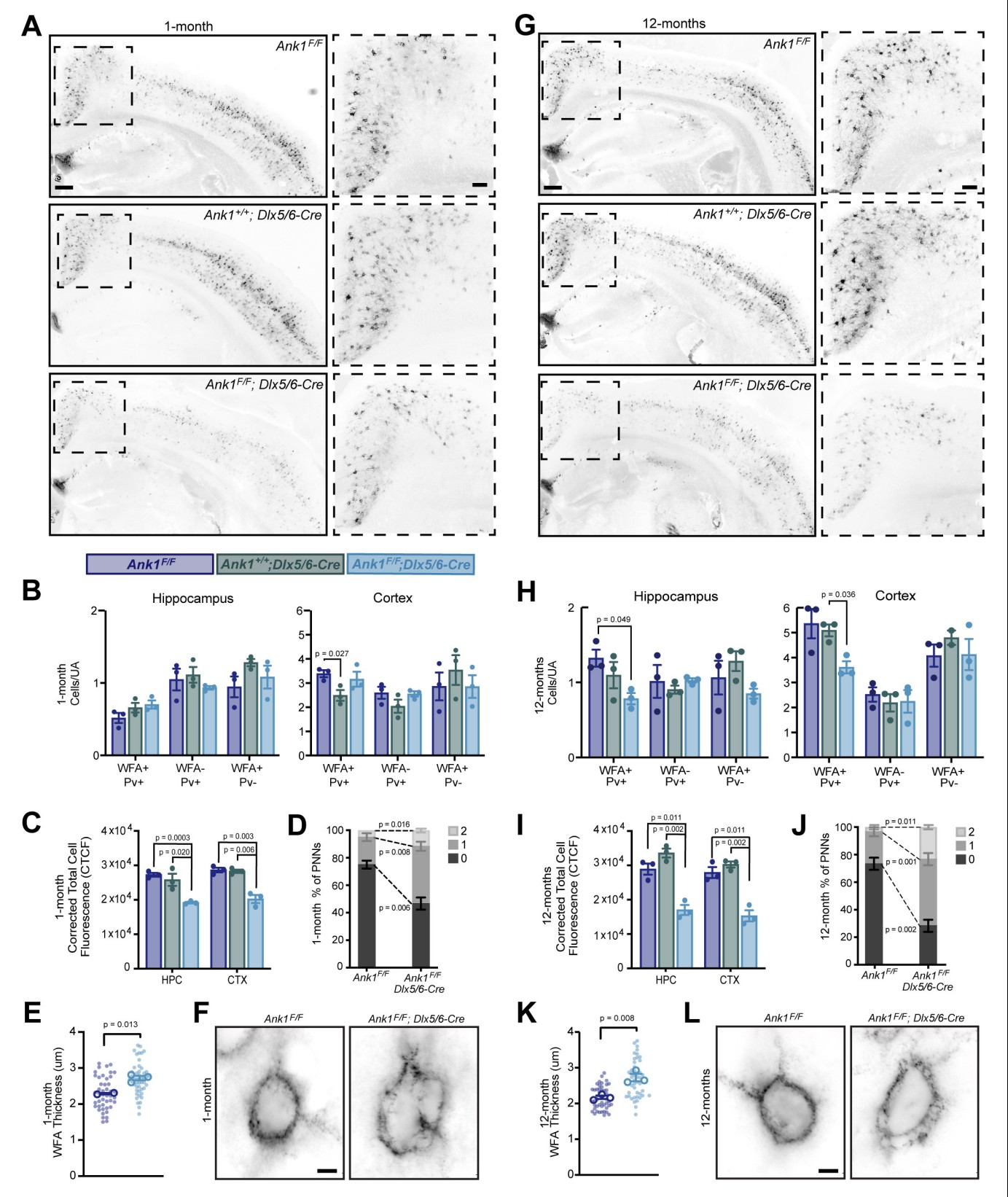

**Figure 3.** AnkR maintains perineuronal net density and structure. (**A**) Fluorescent WFA labeling of PNNs in coronal sections of cortex and hippocampus in one-month-old mice. Genotypes of the respective mice are indicated. Boxed regions are shown to the right. Scalebars, 250 μm and 100 μm. (**B**)
*Figure 3 continued on next page*

*Figure 3 continued*

Quantification of colocalization between WFA and Pv labeling per unit area (UA) in hippocampus and cortex. N=three mice/group. (C) Corrected total cell fluorescence (CTCF) in hippocampus (HPC) and cortex (CTX). N=three mice/group, n=230 cells/animal. (D) Semi-quantitative analysis of high magnification WFA showing increased hole size and disruption of PNNs in retrosplenial (RSP) cortex of 1-month mice. N=three mice/group, n=20 cells/animal. (E) Analysis of high-magnification WFA showing increased thickness of WFA and disruption of PNNs in somatosensory cortex of 1-month mice; *Ank1^{F/F}* (N=two mice; n=45 cells), *Ank1^{F/F}; Dlx5/6-Cre* (N=three mice; n=51 cells). Small solid dots indicate individual cells, large open dots indicate animal means. Error bars indicate mean ± SEM. (F) Fluorescent WFA label of PNNs in RSP cortex of 1-month-old mice. Genotypes of the respective mice are indicated. Scalebars, 5 μm. (G) Fluorescent WFA labeling of PNNs in coronal sections of cortex and hippocampus in 12-month-old mice. Genotypes of the respective mice are indicated. Boxed regions are shown to the right. Scalebars, 250 μm and 100 μm. (H) Quantification of colocalization between WFA and Pv labeling per unit area (UA) in hippocampus and cortex. N=three mice/group. (I) CTCF in HPC and CTX. N=three mice/group, n=230 cells/animal. (J) Semi-quantitative analysis of high magnification WFA showing increased hole size and disruption of PNNs in RSP cortex of 12-month mice. N=three mice/group, n=20 cells/animal. (K) Analysis of high-magnification WFA showing increased thickness of WFA and disruption of PNNs in somatosensory cortex of 12 month mice; *Ank1^{F/F}* (N=three mice; n=46 cells), *Ank1^{F/F}; Dlx5/6-Cre* (N=three mice; n=45 cells). Small solid dots indicate individual cells, large open dots indicate animal means. Error bars indicate mean ± SEM. (L) Fluorescent WFA label of PNNs in RSP cortex of 12-month-old mice. Genotypes of the respective mice are indicated. Scalebars, 5 μm. Error bars indicate mean ± SEM. N=3/group. The online version of this article includes the following source data and figure supplement(s) for figure 3:

**Source data 1.** Source data related to *Figure 3* .
**Figure supplement 1.** Labeling of control and AnkR cKO neurons in cortex using fluorescent WFA and antibodies against AnkR.
**Figure supplement 2.** WFA labeling is reduced across cortical regions.
**Figure supplement 2—source data 1.** Source data related to *Figure 3—figure supplement 2* .

## AnkR influences the intrinsic excitability and firing properties of WFA⁺ fast-spiking interneurons

To determine how loss of AnkR impacts the intrinsic excitability and firing properties of fast-spiking interneurons, we performed whole-cell current clamp recordings on layer 5 PNN⁺ interneurons in somatosensory cortical slices from *Ank1^{F/F}*, *Ank1^{+/+};Dlx5/6-Cre*, and *Ank1^{F/F};Dlx5/6-Cre* mice. Fast-spiking interneurons were identified in live brain slices by labeling PNNs with fluorescent WFA (*Figure 5—figure supplement 1A–C*). Indeed, 64 out of 66 WFA⁺ cells recorded were fast-spiking interneurons and the two non-fast spiking cells were not included in the analysis. We found that the resting membrane potential, input resistance, membrane capacitance, and rheobase current were not significantly altered in *Ank1^{F/F};Dlx5/6-Cre* neurons compared to controls (*Supplementary file 1*). We determined the action potential latency, threshold, amplitude, half-width, afterhyperpolarization (AHP) amplitude, and AHP time from the single action potential elicited by the rheobase current (*Figure 5A*). Loss of AnkR decreased the action potential latency and threshold (*Figure 5B,C*, and

*Supplementary file 1*) without changing the amplitude of the action potential (*Figure 5D*, and *Supplementary file 1*). Interestingly, loss of AnkR also altered the shape of the action potential, resulting in a 47% broader action potential with a shallower and delayed AHP (*Figure 5E–H*, and *Supplementary file 1*). Furthermore, loss of AnkR also resulted in altered phase plane plots and a reduced maximum dV/dt (*Figure 5—figure supplement 1D,E*).

We also recorded trains of action potentials evoked by different levels of current injection (*Figure 5I,J*). At two times, the action potential threshold current, fast-spiking interneurons from *Ank1^{F/F};Dlx5/6-Cre* mice display a decreased firing frequency during the first 100 ms of current injection (*Figure 5K*), but normal spike frequency adaptation (*Figure 5L*). However, the spike amplitude adaptation was enhanced, resulting in a strong reduction in the amplitudes of action potentials toward the end of spike

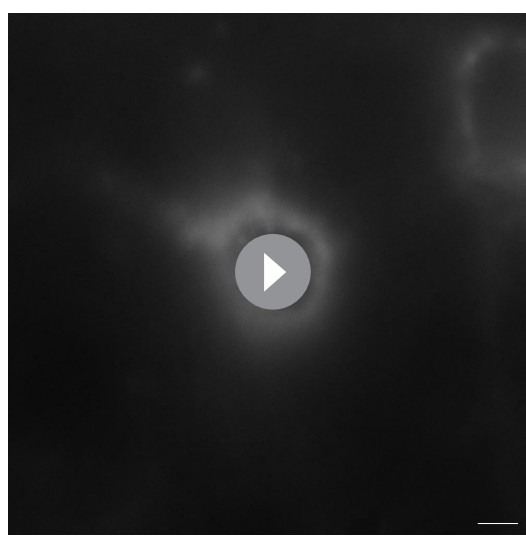

**Video 1.** WFA in 12-month *Ank1^{F/F}*. Scalebar, 5 μm.
https://elifesciences.org/articles/66491#video1

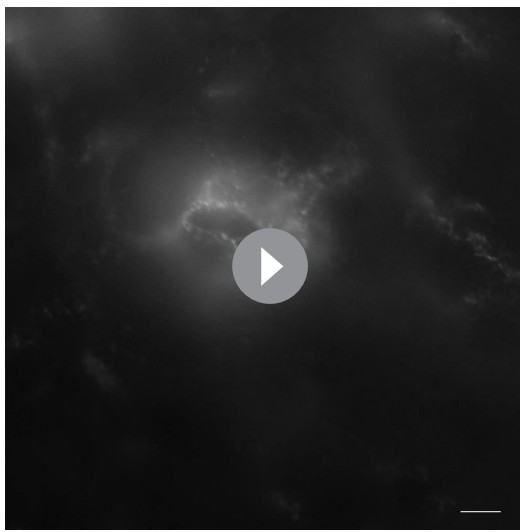

**Video 2.** WFA in 12-month *Ank1^{F/F};Dlx5/6-Cre*. Scalebar, 5 μm.
https://elifesciences.org/articles/66491#video2

trains (*Figure 5I,M*). In fact, *Ank1^{F/F};Dlx5/6-Cre* cells often entered depolarization block at much lower current levels than control cells (*Figure 5I, N*). Thus, the reduction in firing frequency for *Ank1^{F/F};Dlx5/6-Cre* cells at high levels of current injection is actually an underestimation, because many *Ank1^{F/F};Dlx5/6-Cre* cells prematurely reached depolarization block and were not represented by the current-firing frequency curve (*Figure 5J*).

In addition to examining intrinsic excitability of fast-spiking interneurons, we also performed whole-cell voltage clamp recordings to record miniature postsynaptic currents. We observed no significant differences between the controls and *Ank1^{F/F};Dlx5/6-Cre* mice for the frequency and amplitude of miniature excitatory postsynaptic currents (mEPSC) or miniature inhibitory postsynaptic currents (mIPSC), as well as the ratio of excitatory to inhibitory inputs (E/I ratio) (*Figure 5—figure supplement 2A–F*). Taken together, these results show that loss of AnkR alters the intrinsic properties of WFA⁺ fast-spiking inhibitory interneurons. Furthermore, they are consistent with what has been reported in another model where PNNs of fast-spiking interneurons are disrupted due to loss of Bcan (*Favuzzi et al., 2017*); however, loss of Bcan also induced changes in mPSCs. Intriguingly, the results are very similar to those seen in mice with loss of Kv3 K⁺ channel expression in fast-spiking interneurons (*Lau et al., 2000*). Hence, we hypothesize that loss of AnkR may impact Kv3 K⁺ channels.

## AnkR recruits and maintains Kv3.1b K⁺ channels at the neuronal membrane

Among the Kv channels, Kv3.1b is highly expressed in WFA⁺ cortical interneurons (*Härtig et al., 1999*) Kv3.1b is also found at some CNS (central nervous system) nodes of Ranvier, but not AIS (*Devaux et al., 2003*). Similarly, AnkR is enriched in WFA⁺ cortical interneurons (*Figure 2I*), and can be found at some nodes of Ranvier (*Ho et al., 2014*), but not AIS (*Liu et al., 2020*). Based on these similarities, we considered Kv3.1b to be a good candidate to interact with AnkR in somatic membranes of WFA⁺ cortical interneurons. Immunostaining of control 1- and 12-month-old somatosensory cortex showed that AnkR⁺ neurons were also Kv3.1b⁺ (*Figure 6A*, and *Figure 6—figure supplement 1A*), and that AnkR and Kv3.1b colocalize at the neuronal membrane (*Figure 6B*). This same colocalization was also seen in human cortex (*Figure 6C*). Remarkably, AnkR-deficient *Ank1^{F/F}; Dlx5/6-Cre* mice have a profound reduction in Kv3.1b⁺ neurons at both 1- and 12 months of age (*Figure 6A,B*, and *Figure 6—figure supplement 1A*) and nearly complete loss of Kv3.1b immunofluorescence (*Figure 6D*). Compared to controls, *Ank1^{F/F};Dlx5/6-Cre* mice have a ~50% reduction in Kv3.1b protein (*Figure 6E,F*), which persists in 12-month-old mice (*Figure 6—figure supplement 1B,C*). Furthermore, we also observed loss of Kv3.1a and Kv3.2, both of which may be functionally redundant and form heterotetramers with Kv3.1b; immunostaining using antibodies recognizing Kv3.1a/Kv3.1b and Kv3.2 showed a reduction in immunofluorescence at 1 month (*Figure 6—figure supplement 1D–G*). These results show that AnkR is required to maintain clustering of Kv3.1b in the somatic membrane of GABAergic interneurons.

To determine if AnkR and Kv3.1b interact, we co-transfected HEK cells with AnkR-GFP and Flag-tagged Kv3.1b. AnkR-GFP efficiently co-precipitates full-length Flag-Kv3.1b (*Figure 6G*). We then constructed serial C-terminal truncations of Flag-Kv3.1b. We found that AnkR-GFP pulled down amino acids (aa) 1–546 of Kv3.1b (*Figure 6G*). However, additional shortening of the C-terminus blocked the interaction with AnkR. Thus, AnkR binds to the region of Kv3.1b including aa 510–546 (*Figure 6G*). To further define the motif in Kv3.1b that interacts with AnkR, we generated additional C-terminal deletions of just six aa each, spanning aa 510–546 of Kv3.1b. We found that aa 516–522

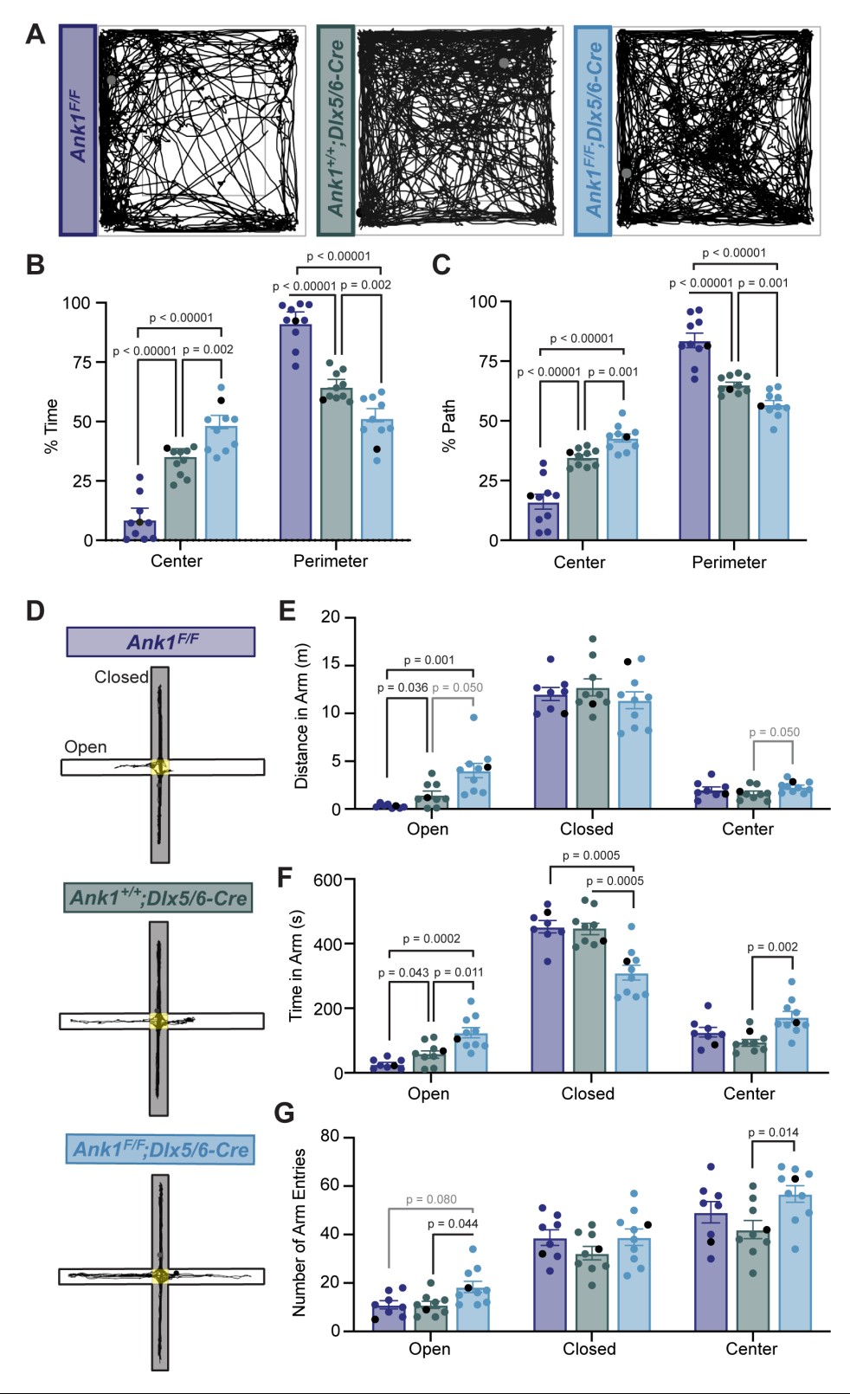

**Figure 4.** Mice lacking AnkR in GABAergic forebrain interneurons have reduced anxiety. (**A**) Thirty min-long recordings of mouse trajectories in the open-field assay. Genotypes are indicated. (**B**) The percent of time spent in the center or perimeter of the open field. (**C**) The percent of the total path spent in the center or perimeter of the open field. (**D**) 10 minute-long recordings of mouse trajectories in the elevated plus maze. (**E**) The distance

*Figure 4 continued on next page*

*Figure 4 continued*

traveled in the open arm, closed arm, or the center of the elevated plus maze. (F) The time spent in the open arm, closed arm, or the center of the elevated plus maze. (G) The number of entries into the open arm, closed arm, or center of the elevated plus maze. In all panels, error bars indicate mean ± SEM. Black circles indicate the animals corresponding to the representative traces.

The online version of this article includes the following figure supplement(s) for figure 4:

**Figure supplement 1.** Quantification of open field and elevated plus maze assays.

---

of Kv3.1b (EDCPHI) are required for AnkR binding (*Figure 6H*). A nearly identical motif is also present in Kv3.3, but not Kv3.2 (*Figure 6I*). Kv3.3 is also highly expressed in Pv$^+$ neurons (*Chang et al., 2007*) and immunostaining of control one-month-old somatosensory cortex showed that a subset of AnkR$^+$ neurons also express Kv3.3 (*Figure 6—figure supplement 1H*). Using brain homogenates, we found that Kv3.1b co-immunoprecipitates AnkR (*Figure 6J*). However, we were unable to detect Kv3.1b after immunoprecipitation of AnkR; this may suggest that only a small fraction of the total AnkR interacts with Kv3.1b. Together, these results show that AnkR binds directly to Kv3.1b.

Since Kv3.1b interacts with AnkR and is required for its membrane localization (*Figure 6A,B*), we next determined if AnkR is sufficient to recruit Kv3.1b to neuronal membrane domains. Although some CNS nodes of Ranvier have clustered Kv3.1b, most peripheral nervous system nodes of Ranvier normally have high levels of Kv7.2/3 K$^+$ channels rather than Kv3.1b (*Figure 6K,L*; *Pan et al., 2006*). Kv7.2 K$^+$ channel clustering requires binding to AnkG since Kv7.2 is absent from nodes in the ventral roots of AnkG-deficient (*Ank3$^{F/F}$;Chat-Cre*) mice (*Figure 6K,L*). AnkR clusters nodal Na$^+$ channels (*Ho et al., 2014*) and neurofascin 186 in the ventral roots of *Ank3$^{F/F}$;Chat-Cre* mice (*Figure 6K*). Although very few nodes in ventral root normally have Kv3.1b (*Figure 6L*), the replacement of AnkG by AnkR in *Ank3$^{F/F}$;Chat-Cre* mice is sufficient to recruit and cluster Kv3.1b to nearly all nodes (*Figure 6K,L*); and nodes in control spinal cord that have high levels of AnkR also have clustered Kv3.1b (*Figure 6—figure supplement 1I*). Similarly, AnkR recruits Kv3.3 K$^+$ channels to nodes in AnkG-deficient ventral root axons (*Figure 6—figure supplement 1J*). Together, these results show that although AnkG and AnkR can both cluster Na$^+$ channels at nodes of Ranvier, AnkG preferentially clusters Kv7.2/3 K$^+$ channels and links them to the cytoskeleton through α2/β4 spectrin (*Huang et al., 2017b*), while AnkR is both necessary and sufficient to recruit Kv3.1b/3 K$^+$ channels to neuronal membranes and nodes of Ranvier, and links them to the cytoskeleton through α2/β1 spectrin (*Ho et al., 2014*; *Huang et al., 2017a*; *Figure 6—figure supplement 1K*). Thus, the type of K$^+$ channel found at nodes of Ranvier is dictated by ankyrins.

## Discussion

Ankyrins are well-known to function in neurons as scaffolds that link ion channels and membrane proteins to the spectrin cytoskeleton (*Bennett and Lorenzo, 2013*). We previously showed that in the absence of AnkG, AnkR can function at nodes of Ranvier as a secondary Na$^+$ channel clustering mechanism (*Ho et al., 2014*). The rescue depends on AnkR's recruitment to nodes from a pre-existing, unclustered pool. These findings motivated us to determine AnkR's normal functions in the nervous system since it is unlikely that AnkR functions only as a backup for nodal Na$^+$ channel clustering, since pathogenic *ANK1* variants are associated with nervous system dysfunction, and since altered methylation of *ANK1* is associated with Alzheimer's disease. Our experiments confirm that in general, AnkR acts as a scaffolding protein like AnkB and AnkG. However, unlike AnkB and AnkG which are broadly expressed in all neurons, AnkR is highly and specifically enriched in subsets of neurons, including fast-spiking GABAergic interneurons, where it assembles and stabilizes unique protein complexes necessary for the proper function of these cells (*Figure 7*).

### AnkR maintains PNNs

GABAergic interneurons are surrounded by PNNs, and our proteomics experiments revealed that AnkR indirectly interacts with PNN components (*Figure 7*). PNNs are proposed to have many functions including regulation of synaptic plasticity, excitation and inhibition, ion buffering, and even protection against neurodegeneration and neurotoxicity (*Cabungcal et al., 2013*; *Fawcett et al., 2019*;

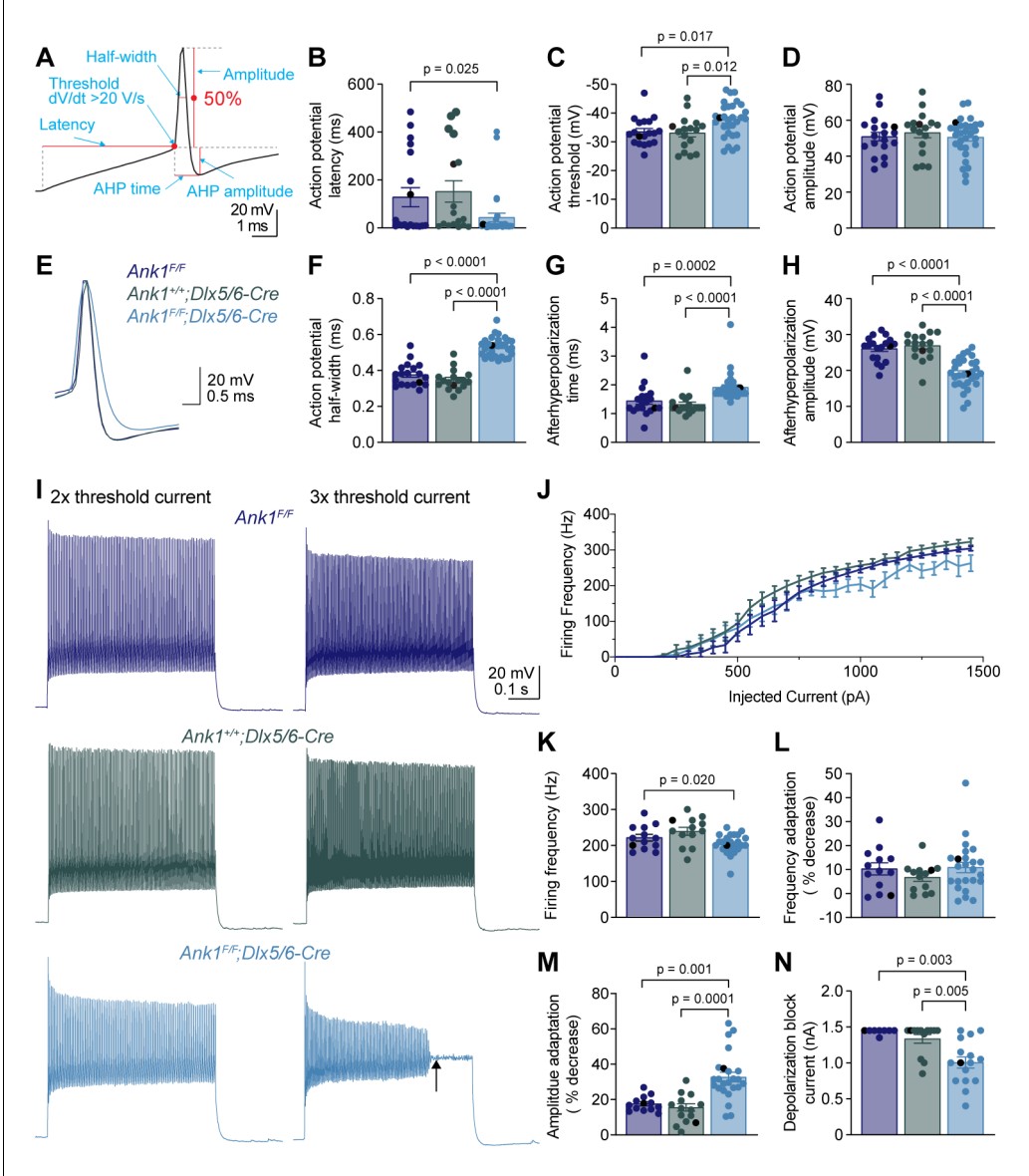

**Figure 5.** WFA[+] neurons in *Ank1^{F/F};Dlx5/6-Cre* mice have abnormal action potentials and spike train characteristics. (**A**) A representative action potential illustrating the measurement of action potential parameters. AHP, afterhyperpolarization. (**B–D**) Summary data showing the action potential latency (**B**), action potential threshold (**C**), and action potential amplitude (**D**) from the single action potential evoked by rheobase current. (**E**) Representative single action potentials evoked by rheobase currents from *Ank1^{F/F}*, *Ank1^{+/+};Dlx5/6-Cre*, and *Ank1^{F/F};Dlx5/6-Cre* cells. Action potentials are aligned at 50% of the rising phase on X axis and peak on Y axis. Note the wider action potential with shallower and delayed afterhyperpolarization in *Ank1^{F/F};Dlx5/6-Cre* cell. (**F–H**) Summary data showing the action potential half-width (**F**), afterhyperpolarization time (**G**), and afterhyperpolarization amplitude (**H**) from the single action potentials evoked by rheobase currents. (**I**) Representative spike trains from *Ank1^{f/f}*, *Ank1^{+/+};Dlx5/6-Cre*, and *Ank1^{F/F};Dlx5/6-Cre* cells in response to 500 ms current injection. Left and right traces show the spike trains evoked by currents that are two and three times of the action potential threshold currents, respectively. Note the strong amplitude adaptation and premature depolarization block indicated by the arrow in the right trace of *Ank1^{F/F};Dlx5/6-Cre* cell. (**J**) The average firing frequency during 500 ms current injection as a function of injected currents. Note, recording was stopped at maximal 1450 pA current or when cells reached depolarization block. Since 13 out of 15 *Ank1^{f/f};Dlx5/6-Cre+* cells reached depolarization block prior to 1450 pA current while only 3 out of 19 control cells reached depolarization block prior to 1450 pA, the firing frequency is overestimated in the high current range for *Ank1^{F/F};Dlx5/6-Cre* neurons. (**K–M**) Summary data showing the average firing frequency during the first 100 ms (**K**), spike frequency adaptation (**L**), and amplitude adaptation (**M**)

*Figure 5 continued on next page*

*Figure 5 continued*

from the spike trains evoked by currents that are two times of the action potential threshold currents. (**N**) Summary data showing the minimal currents that caused the cells to enter depolarization block. If the maximal injected current (1450 pA) did not cause depolarization block then 1450 pA was recorded as the result. For all panels, each circle represents one neuron and the black circles indicate the representative cells in (**E and I**). Bar graphs represent mean ± SEM. Statistical significance was determined by one-way ANOVA or Kruskal-Wallis test with multiple comparisons.

The online version of this article includes the following source data and figure supplement(s) for figure 5:

**Figure supplement 1.** Flourescein-WFA labeling of perineuronal nets in live slices for electrophysiology.

**Figure supplement 1—source data 1.** Source data related to *Figure 5—figure supplement 1* .

**Figure supplement 2.** Analysis of miniature excitatory postsynaptic currents.

**Figure supplement 2—source data 1.** Source data related to *Figure 5—figure supplement 2* .

*Suttkus et al., 2016*). The connection between AnkR and PNNs is remarkably similar to the connection between AnkG and perinodal and AIS ECMs. These latter ECMs interact with AnkG through the CAM neurofascin 186, and loss of either AnkG or neurofascin 186 blocks their assembly (*Amor et al., 2017*; *Hedstrom et al., 2007*; *Susuki et al., 2013*). In contrast, the membrane receptors and mechanisms of PNN assembly and maintenance are unknown. Our proteomics experiments revealed candidates and suggest that NrCAM and PlexinA4, together with Nrp1, may participate in assembly or maintenance of PNNs. These CAMs co-immunoprecipitate with AnkR. NrCAM and Nrp1 colocalize with a subset of AnkR$^+$ neurons. Furthermore, loss of AnkR dramatically reduces the number of perisomatic NrCAM$^+$/WFA$^+$ neurons, suggesting that as for PNNs, AnkR is required to maintain perisomatic NrCAM in WFA$^+$ neurons. Future studies of PNNs in NrCAM and PlexinA4-deficient mice may help to determine the role of these CAMs in PNN assembly and maintenance. In addition, other interesting candidates identified in our proteomics experiments may also function as receptors. For example, we identified the adhesion G-protein-coupled receptor Gpr158 and the CAM Limbic System Associated Membrane Protein (Lsamp). It will be interesting and important to determine if these membrane proteins also function with AnkR to assemble, modulate, or maintain PNNs, and we speculate that multiple receptors link AnkR to PNNs.

We observed the disruption of the compact PNN structure and a ~50% reduction in WFA fluorescence intensity in the absence of AnkR in juvenile and adult mice. This is notable since mice lacking four PNN components (TnC, TnR, and the chondroitin sulfate proteoglycans Bcan and neurocan) have a significant reduction in PNN structure, area, and WFA fluorescence during development, but PNNs normalize by postnatal day 35 (*Gottschling et al., 2019*). Thus, loss of AnkR has more profound effects on maintenance of PNNs than even removing components of the PNNs themselves. This suggests that although AnkR is not necessary for the assembly of PNNs, the receptors responsible for PNN maintenance converge on AnkR (*Figure 7*). Nevertheless, since PNNs still assemble in the absence of AnkR, other AnkR-independent PNN assembly mechanisms must also exist.

## AnkR, PNNs, and neurological disease

Mice lacking AnkR in GABAergic interneurons showed reduced anxiety in both the open field and elevated plus maze tests. Increased anxiety has been correlated with increased PNN density (*Murthy et al., 2019*). Conversely, the selective serotonin reuptake inhibitor (SSRI) fluoxetine reduces anxiety as well as PNN density (*Ohira et al., 2013*). The results of these studies are consistent with our observation that *Ank1$^{F/F}$;Dlx5/6-Cre* mice have both reduced anxiety and a significant reduction in PNN density (*Figure 7*). Some studies have reported large reductions in PNNs throughout the brains of patients with schizophrenia (*Berretta, 2012*; *Mauney et al., 2013*). Multiple genome wide association studies (GWAS) implicate *ANK1* as a schizophrenia-associated gene (*Aberg et al., 2013*; *Fromer et al., 2014*) Schizophrenia Working Group of the *Schizophrenia Working Group of the Psychiatric Genomics Consortium, 2014*, and several of the top candidates for AnkR-interacting proteins identified in our mass spectrometry analysis (e.g. *Myh11*, *Kif1a*, *Itsn1*, and *Slc25a12*) are also schizophrenia-associated genes (*Fromer et al., 2014*) Schizophrenia Working Group of the *Schizophrenia Working Group of the Psychiatric Genomics Consortium, 2014* Kv3.1b is also reduced in patients with schizophrenia (*Yanagi et al., 2014*). A number of epigenome-wide association studies in Alzheimer's disease patients consistently report

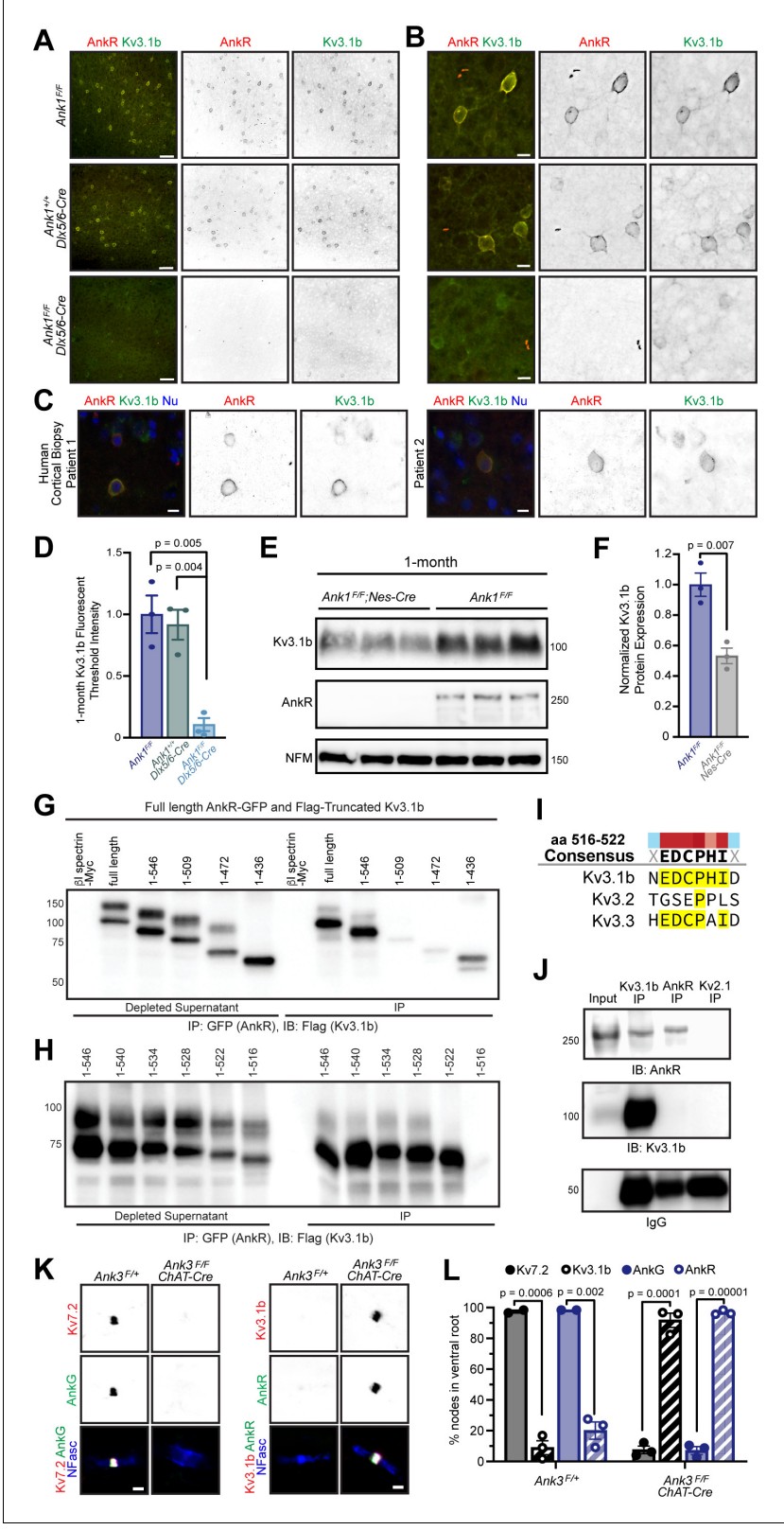

**Figure 6.** AnkR binds to Kv3.1b K$^+$ channels and is both necessary and sufficient for its membrane localization and clustering. (**A, B**) Immunostaining of 1-month-old somatosensory cortex for AnkR (red) and Kv3.1b (green). Low-magnification images are shown in (**A**) and high-magnification images in (**B**). The genotypes analyzed are shown. Scalebars, 50 μm in (**A**) and 10 μm in (**B**). (**C**) Immunostaining of human cortical biopsies from two separate

*Figure 6 continued on next page*

*Figure 6 continued*

patients using antibodies against AnkR (red) and Kv3.1b (green), and DAPI (blue) to label nuclei (Nu). Scalebars, 10 μm. (D) Quantification of Kv3.1b immunofluorescence intensity in control and *Ank1$^{F/F}$;Dlx5/6-Cre* mice. Error bars indicate mean ± SEM. N=3/group. (E) Immunoblots of brain homogenates from three one-month-old control and three 1-month-old AnkR-deficient brains using antibodies against Kv3.1b, AnkR, and NFM. (F) Quantification of Kv3.1b protein normalized to NFM. (G, H). Immunoblots of AnkR-GFP immunoprecipitations in cells co-expressing AnkR-GFP with Myc-tagged β1 spectrin, full length Flag-tagged Kv3.1b, or truncated versions of Flag-tagged Kv3.1b. The amino acids included in the Flag-tagged Kv3.1b truncation mutants are indicated. (I) The consensus AnkR-binding motif present in Kv3.1b and Kv3.3, but not Kv3.2. (J) Immunoblots of Kv3.1b, AnkR, and IgG immunoprecipitation reactions using antibodies against AnkR and Kv3.1b. (K) Immunostaining of ventral root nodes of Ranvier in *Ank3$^{F/+}$ and Ank3$^{F/F}$;Chat-Cre* mice using antibodies against AnkG (green), Kv7.2 (red), and neurofascin (NFasc, blue) on the left, and AnkR (green), Kv3.1b (red), and NFasc (blue) on the right. Scalebars, 1 μm. (L) Quantification of the percentage of nodes of Ranvier labeled for Kv7.2, Kv3.1b, AnkG, and AnkR in *Ank3$^{F/+}$* and *Ank3$^{F/F}$;Chat-Cre* mice. *Ank3$^{F/+}$* - Kv7 (N=two mice; n=217 nodes), Kv3.1b (N=three mice; n=301 nodes), AnkG (N=two mice; n=222 nodes), AnkR (N=three mice; n=286 nodes). *Ank3$^{F/F}$; Chat-Cre* - Kv7 (N=three mice; n=252 nodes), Kv3.1b (N=three mice; n=244 nodes), AnkG (N=three mice; n=251 nodes), AnkR (N=three mice; n=249 nodes). Error bars indicate mean ± SEM.

The online version of this article includes the following source data and figure supplement(s) for figure 6:

**Figure supplement 1.** Kv3.1b membrane localization requires AnkR.

**Figure supplement 1—source data 1.** Source data related to *Figure 6—figure supplement 1*.

---

neuropathology-associated DNA hypermethylation of *ANK1* (*De Jager et al., 2014*; *Higham et al., 2019*; *Lunnon et al., 2014*; *Smith et al., 2019a*; *Smith et al., 2019b*). However, the consequence of this hypermethylation for AnkR protein expression is unknown. Experiments in APP/PS1 mice also report significant reductions in Kv3.1b (*Boda et al., 2012*). Since Kv3.1b levels in GABAergic neurons depend on AnkR, reduced expression of AnkR in the Alzheimer's disease brain could result in altered neuronal excitability or circuit function due to the decreased levels of Kv3.1b. Similarly, PNN density has been reported to be reduced in both human Alzheimer's disease brains and brains from the 5xFAD mouse model of Alzheimer's disease (*Crapser et al., 2020*). Thus, dysregulation of *ANK1*, leading to reduced AnkR, Kv3.1b, and PNN density, may be a common pathomechanism in neurological and psychiatric disease. Intriguingly, the expression of Kv3.1b in schizophrenic patients is corrected by antipsychotic drugs (*Yanagi et al., 2014*). It will be interesting to determine if antipsychotic drugs similarly affect AnkR protein levels or PNNs.

## AnkR recruits Kv3.1b to the neuronal membrane

Loss of AnkR from GABAergic interneurons significantly altered their intrinsic and firing properties, suggesting disrupted K$^+$ channel function. However, the changes in excitability may also reflect loss of both Kv3.1b K$^+$ channels and reduced PNNs, since Bcan-deficient mice have similar changes in the excitability of their GABAergic interneurons (*Favuzzi et al., 2017*). We found that AnkR interacts directly with Kv3.1b and is required to maintain Kv3.1b in GABAergic neurons. We localized Kv3.1b's AnkR-binding motif to six residues in its C-terminus: EDCPHI. This motif is different than the previously characterized pan ankyrin-binding motifs present in all Na$^+$ channels (*Garrido et al., 2003*) and L1CAMs (*Garver et al., 1997*; *Tuvia et al., 1997*). In contrast, there is specificity among the ankyrin-binding capacities of different K$^+$ channels: AnkR interacts with Kv3.1b and is both necessary and sufficient to induce Kv3.1b clustering in the soma of GABAergic interneurons and a subset of CNS nodes of Ranvier, while AnkG binds Kv7.2/3 and is necessary and sufficient for its clustering at AIS and nodes of Ranvier. Although the Kv7.2/3 motif is highly homologous to the Na$^+$ channel ankyrin-binding motif, AnkR does not cluster Kv7.2/3 (*Wang et al., 2018*). When Kv3.1b is present at nodes, it is associated with AnkR. A nearly identical motif is also present in Kv3.3. Like Kv3.1b, Kv3.3 is enriched in Pv$^+$ interneurons in the forebrain, hippocampus, deep cerebellar nuclei, and Purkinje neurons in the cerebellum (*Chang et al., 2007*); these same cells also express high levels of AnkR (*Kordeli and Bennett, 1991*). Although one previous study suggested Kv3.3 is not found at nodes (*Chang et al., 2007*), our results show that CNS nodes with AnkR have both Kv3.1b and Kv3.3, most likely as heterotetramers. Thus, K$^+$ channel diversity among nodes of Ranvier is dictated by the specific ankyrin scaffolds present at nodes.

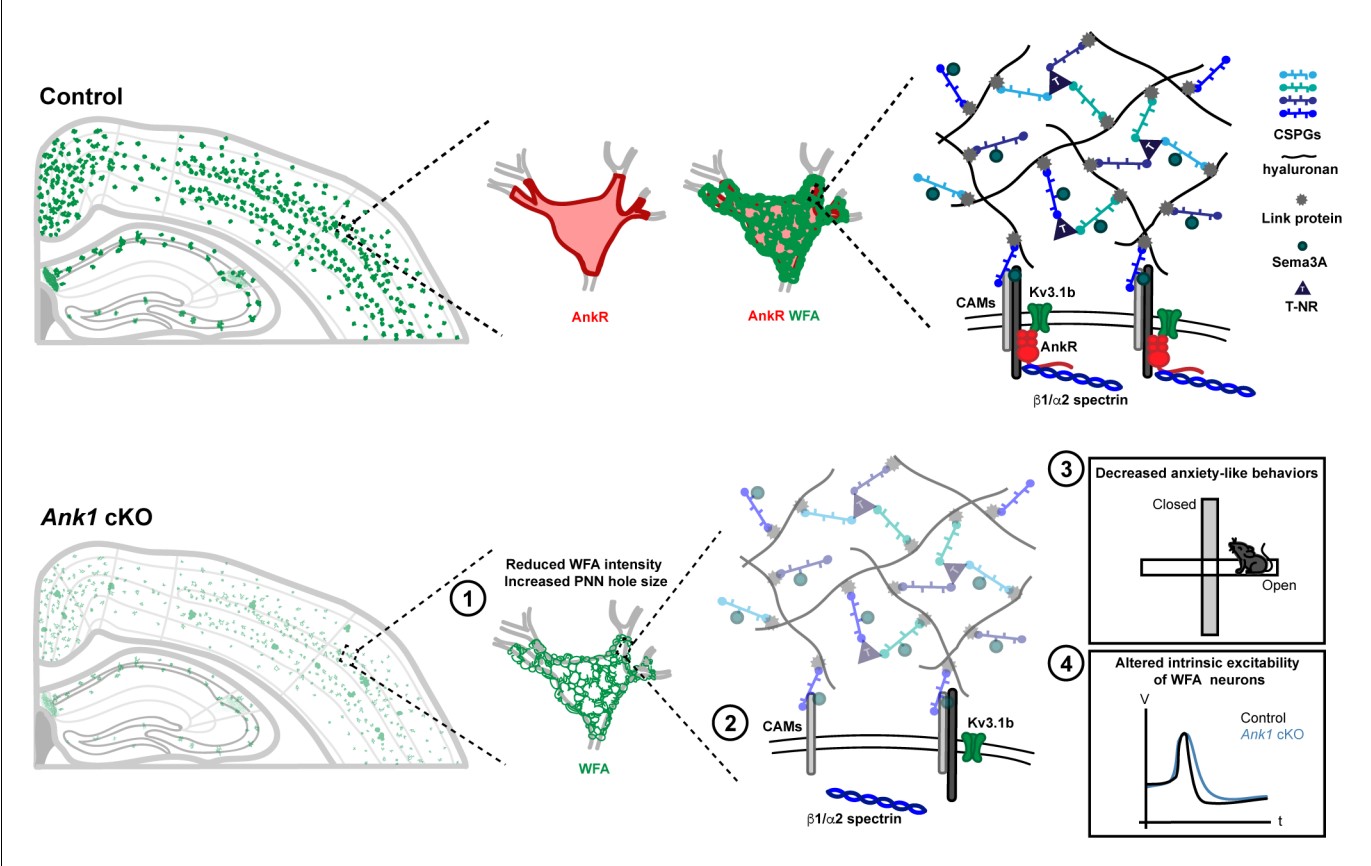

**Figure 7.** AnkR is highly expressed in the perisomatic region of Pv[+] interneurons which are surrounded by PNNs, a specialized ECM structure. AnkR is a scaffolding protein that binds to and stabilizes PNN-associated CAMs (including NrCAM and PlexinA4) and ion channels (including Kv3.1b) by linking them to the β1-α2 spectrin-based cytoskeleton. Loss of AnkR results in (**1**) altered PNN morphology including reduced WFA intensity and decreased compactness of the nets; (**2**) molecular changes including reduced β1 spectrin, PNN-associated NrCAM, and Kv3.1b; (**3**) behavioral changes including decreased anxiety-like behaviors in the open field and elevated plus maze; and (**4**) electrophysiological changes including decreased AP latency and threshold, broader APs with shallower and delayed AHP, and decreased firing rate during current injection.

Since mutations in AnkR cause severe hemolytic anemia, its role in the nervous system has mostly been ignored. However, our results demonstrate that AnkR is much more than just red blood cell ankyrin. Our studies reveal AnkR's critical role in fast-spiking GABAergic interneurons, its interacting proteins, and show that AnkR is necessary for normal interneuron function.

## Materials and methods

### Animals

AnkR conditional knockout mice were generated using cell-type-specific Cre-mediated deletion of the *Ank1* gene. The targeting construct was designed by introducing *loxP* sites flanking exons 26 and 27 of *Ank1*, upstream of the spectrin binding domain. The Cre-mediated removal of these exons will cause a frame-shift mutation resulting in a premature stop codon in exon 28. Forward primer: 5'- GGG AAA CTC CAC AGA GCC TGA CGG GTC AGT-3', Reverse primer: 5'- GGC GTC CCT ATG TTC CAT CCT ATA GAT GAC T-3'. Generation of the target construct, ES cell electroporation, colony selection, blastocyst injection, and generation of chimeric mice were a service of the University of Rochester Medical Center Transgenic Core Facility. The strategy is similar to that successfully used to create the *Ank2* and *Ank3* floxed mice where *loxP* sites flank exons 23/24 and 22/23, respectively (*Chang et al., 2014*; *Ho et al., 2014*).

*Ank1*[F/F] mice (IMSR Cat# JAX:036512) were generated and backcrossed to C57BL/6 (IMSR Cat# JAX:000664, RRID:IMSR_JAX:000664) for at least four generations before being crossed to *Nestin-*

*Cre* transgenic mice (IMSR Cat# JAX:003771, RRID:IMSR_JAX:003771), *Dlx5/6-Cre* transgenic mice (IMSR Cat# JAX:008199, RRID:IMSR_JAX:008199). *Ank1^{pale/pale}* mice were obtained from Jackson Laboratory (IMSR Cat# JAX:009157, RRID:IMSR_JAX:009157). Since germline recombination has been found to occur in these Cre lines (*Luo et al., 2020*), immunostaining using anti-AnkR antibodies was used as a secondary confirmation of genotype. *Ank3^{F/F}* mice (IMSR Cat# JAX:029797, RRID: IMSR_JAX:029797) were crossed with *Chat-Cre* transgenic mice (IMSR Cat# JAX:006410, RRID: IMSR_JAX:006410).

Both male and female mice were used in our studies. All experiments were conducted in compliance with the National Institutes of Health Guide for the Care and Use of Laboratory Animals and were approved by the Animal Care and Use Committee at Baylor College of Medicine.

## Blood transfusion and bone marrow transplant in *Ank1^{pale/pale}* animals

The *pale* mutation in *Ank1* results in severe anemia characterized by pale skin tone at birth and death within 1–2 weeks of age (*Ho et al., 2014*). We performed a blood transfusion at P1 and subsequent bone marrow transplant at P30 which enabled *Ank1^{pale/pale}* animals to survive into adulthood. Animals received a 5 mL/kg body weight external jugular vein blood transfusion at P1, and 5 mL/kg bone marrow transplant into the tail vein at P30. Blood was collected from the male breeding mouse and washed with two-times volume of saline prior centrifugated at 700Xg for 20 min at 4°C to remove excess plasma and debris. Bone marrow was isolated from the tibia and femur of close genealogical donor mice. In brief, bones were extracted and placed in 3 mL 2% fetal bovine serum (FBS) HBSS (without $Ca^{2+}$ and $Mg^{2+}$). Marrow was exposed by cutting with sterile scissors and a 25 gauge needle and 3 mL syringe were used to expunge the marrow into a sterile 6 cm dish with 2% FBS HBSS on ice. Cells were then filtered through 1 $cm^2$ 100 µm nylon mesh into a 5 mL FACS tube. Cells were counted and diluted to an injection concentration of 3 million cells/100 µL volume. Throughout life, all animals were closely monitored for changes in weight and behavior.

## Behavioral tests

All behavioral experiments were performed by the same person, in the same undisrupted room, at the same time of day. Mice aged to 9–14 weeks were handled for 3 days prior to the start of testing. Animals began with the open-field assessment followed by the elevated plus maze to assess locomotor activity and anxiety. The open field assessment was conducted in white acrylic open-top boxes (46 x 46 x 38 cm) in a room lit by indirect white light for 30 min. Following this, animals were given 2–3 hr rest in their home cages. The elevated plus maze assessment was then conducted on an elevated platform for 10 min. Activity for these tasks was recorded and analyzed using the ANY-maze Video Tracking System version 4.99 v (Stoelting Co, Wood Dale, IL).

## Antibodies

The primary antibodies used here include: mouse monoclonal antibodies against AnkR (UC Davis/NIH NeuroMab Facility Cat# 75–380, RRID:AB_2491109), β1 spectrin (UC Davis/NIH NeuroMab Facility Cat# 73–374, RRID:AB_2315814), AnkG (UC Davis/NIH NeuroMab Facility Cat# 73–146, RRID:AB_10697718), parvalbumin (UC Davis/NIH NeuroMab Facility Cat# 73–455, RRID:AB_2629420), actin (Millipore Cat# MAB1501, RRID:AB_2223041), tenascinR (R and D Systems Cat# MAB1624, RRID:AB_2207001), aggrecan (Millipore Cat# AB1031, RRID:AB_90460), brevican (UC Davis/NIH NeuroMab Facility Cat# 75–294, RRID:AB_2315824), NrCAM (R and D Systems Cat# MAB2034, RRID:AB_2267411), Kv3.1b (UC Davis/NIH NeuroMab Facility Cat# N16B/8, RRID:AB_2750730), Kv3.2 (UC Davis/NIH NeuroMab Facility Cat# N410/17, RRID:AB_2877619), Kv3.3 (Antibodies-Online Cat# ABIN572016, RRID:AB_10782137), Kv7.2 (James Trimmer, University of California at Davis Cat# N26A/23, RRID:AB_2750761), PanNav (UC Davis/NIH NeuroMab Facility Cat# N419/78, RRID:AB_2877588) Flag-tag or DDDDK-tag (MBL International Cat# M185-3L, RRID:AB_11123930).

Rabbit polyclonal antibodies against AnkR (*Ho et al., 2014*) (RRID:AB_2833096), Ank1 (Thermo Fisher Scientific Cat# PA5-63372, RRID:AB_2638015), neurofilament M (Millipore Cat# AB1987, RRID:AB_91201), parvalbumin (Novus Cat# NB120-11427, RRID:AB_791498), somatostatin (Peninsula Laboratories Cat# T-4103.0050, RRID:AB_518614), versican (Millipore Cat# AB1032, RRID:AB_11213831), PlexinA4 (Abcam Cat# ab39350, RRID:AB_944890), neuropilin-1 (GeneTex Cat#

GTX16786, RRID:AB_422398), Kv3.1 (LSBio (LifeSpan), Cat# LS-C322374, RRID:AB_2891125), Kv3.1b (Alomone Labs Cat# APC-014, RRID:AB_2040166), Kv3.3 (Alomone Labs Cat# APC-102, RRID:AB_2040170), GFP (Thermo Fisher Scientific, Cat# A-11122, RRID:AB_221569).

Chicken polyclonal antibody against Neurofascin (R and D Systems Cat# AF3235, RRID:AB_10890736). Wisteria Floribunda lectins used were: Fluorescein labeled (Vector Laboratories Cat# FL-1351, RRID:AB_2336875 and Bioworld Cat# 21761065–1, RRID:AB_2833087), and Texas-Red (EY Laboratories Cat# F-3101–1, RRID:AB_2315605).

Secondary antibodies were purchased from Jackson ImmunoResearch Laboratories or Life Technologies (IgG specific mouse antibodies and HRP) and were used at 1:1000. The antibody table in extended data provides further antibody information, including knockout validation and example dilutions for the above antibodies; however, we strongly encourage researchers to determine the optimal antibody dilutions for themselves as varying tissue treatment can affect staining conditions.

## Immunofluorescence

Animals were transcardially saline perfused to reduce red blood cells, then brains, spinal cords and roots were dissected and fixed in 4% paraformaldehyde (1 hr for brains and 30 min for spinal cords and roots) on ice and subsequently immersed in 20% sucrose overnight at 4˚C. Tissue was embedded in Tissue-Tek OCT (Sakura Finetek 4583) mounting medium, and frozen on dry ice. Brains were sectioned at 25 µm thickness, ventral roots at 14 µm thickness, and spinal cords were sectioned at 18 µm thickness using a cryostat (Thermo Fisher Scientific Cryostar NX70). Sections were placed on 1% bovine gelatin precoated coverslips (Thermo Fisher Scientific). Sections were blocked with 10% normal goat serum in 0.1M phosphate buffer (PB) with 0.3% Triton X-100 for 1 hr at room temperature. Primary antibodies diluted in the blocking buffer and incubated at room temperature overnight, then sections were washed with blocking buffer. Secondary antibodies and lectins were incubated at room temperature for 2 hr and washed with 0.1M PB.

For immunostaining of human tissues, sections were deparaffinized and rehydrated through a series of alcohols and water. Heat-based antigen retrieval was performed using 1× antigen retrieval solution at pH 9 (Agilent Technologies; Santa Clara, CA) for 1 hr (30 min at 95C, followed by 30 min on ice). Washes with fresh phosphate-buffered saline with Tween 20 (PBS-T) were then performed and PBS with 0.3% Triton X-100 (Sigma, T8787) was applied for 10 min. After additional washes with PBS-T, slides were blocked with 2.5% horse serum (Vector Laboratories, Burlingame, CA) with 1% Tween 20 (ThermoFisher, BP337) and 0.1% BSA in PBS (Thermo Scientific, 37525). Primary antibody along with the above blocking solution was applied for 1–2 hr at room temperature. Secondary antibodies were applied for 1 hr at room temperature, including Alexa Fluor 555 Anti-Rabbit IgG (1:200; A21429), Alexa Fluor 555 Anti-Mouse IgG (1:200; A32727), Alexa Fluor 488 anti-Mouse IgG (1:200; A11001), and Alexa Fluor 488 anti-Rabbit IgG (1:200; A11034) (Alexa Fluor products of Thermo-Fisher), as appropriate. For double-labeling studies, dilutions of both primary and secondary antibodies were combined in 2.5% horse serum and applied. Slides were mounted using Vectashield Antifade mounting medium with 4′,6-diamidino-2-phenylindole (DAPI; Vector Laboratories). These studies were carried out with IRB approval from Houston Methodist Hospital (Pro00010377).

## Image analysis

Immunofluorescence labeling was visualized and images were collected on an AxioImager (Carl Zeiss) fitted with an apotome for optical sectioning, and a digital camera (AxioCam; Carl Zeiss). AxioVision (Carl Zeiss) acquisition software was used for collection of images. Images were also collected using a Nikon Eclipse Ni-E microscope fitted with a motorized X-Y stage for acquisition of fields. Stitching of images was performed using NIS-Elements (Nikon). In some instances, linear contrast and brightness adjustments were performed using Adobe Photoshop, or Z-stacks, 3D reconstruction, and cell counts were performed using NIH FIJI/ImageJ. No other processing of the images was performed.

Human immunofluorescence preparations were reviewed by a neuropathologist (MDC). Images were captured in cellSens software 1.13 (Olympus America, Inc.; Center Valley, PA) on an Olympus BX-43 Microscope using a DP71 camera, an enhanced green fluorescent protein (EGFP) FITC/Cy2 filter cube (set number 49002, Olympus; Center Valley, PA), and a CY3/tetramethylrhodamine-isothiocyanate (TRITC) filter cube (set number 49004, Olympus). To examine the intensity and specificity of

antibody labeling, slides were first examined separately under DAPI, TRITC, and FITC filters, photographed, and then merged in cellSens. All figures were assembled using Adobe Illustrator.

## Image quantification

AnkR, Pv, and SST (*Figure 1C,D*) or WFA and Pv (*Figure 3B,G*) positive cells in cortex and hippocampus were counted manually using the NIH FIJI/ImageJ cell counter. If there was no detectable immunofluorescence, the cell was determined to be absent of AnkR, WFA, Pv, or SST. Colocalization was determined by overlaying channel images and cell count markers, and further confirmed using the X,Y coordinates of count markers. Since AnkR is highly expressed in nearly all Pv (*Figure 1C,D*) and WFA (*Figure 2I*) expressing cells in the regions assessed, all cells expressing Pv and WFA were used for this analysis.

Perisomatic NrCAM (*Figure 2J*), or WFA (*Figure 3A,F*; *Figure 3—figure supplement 2A,B*) intensity were quantified as previously described (*Gottschling et al., 2019*; *McCloy et al., 2014*). In brief, using raw images, NIH FIJI/ImageJ was used to draw an outline around each cell and area, mean fluorescence, integrated density, along with several adjacent background readings were measured. The corrected total cell fluorescence (CTCF) = integrated density – (area of selected cell × mean fluorescence of background readings), was calculated. WFA measurements were averaged within and across multiple brain regions. NIH FIJI/ImageJ was used for a linescan to measure AIS NrCAM (*Figure 2J*) intensity. WFA expression was used as a marker to identify cells with perisomatic NrCAM. Perisomatic and AIS NrCAM fluorescence intensity was normalized by setting the mean of the control group to 1. Semi-quantitative WFA hole grading was performed during acquisition using the Nikon NIS-Elements software by observing cells in the z-plane (*Figure 3D,J*). Multiple WFA thickness measurements were taken for each cell based on saturation profiles of z-stack images using Zeiss AxioVision software (*Figure 3E,K*).

NIH FIJI/ImageJ was used to set a signal-to-noise threshold based on the control group and mean fluorescent intensity was measured in images of somatosensory cortex for Kv3.1b (*Figure 6D*), Kv3.1 (*Figure 6—figure supplement 1F*), and Kv3.2 (*Figure 6—figure supplement 1G*). Fluorescence intensity was normalized by setting the mean fluorescence of the $Ank1^{F/F}$ control group to 1. Thresholds and example images are provided in the extended data. Nodes in ventral roots (*Figure 6L*) were counted manually. Nodes were determined to have AnkG, AnkR, Kv3.1b, or Kv7.2 if there was immunofluorescence at the nodes. If there was not immunofluorescence, the node was determined to be absent of AnkG, AnkR, Kv3.1b, or Kv7.2.

## Immunoblotting

Saline perfused mouse brains were homogenized in homogenization buffer (0.32M sucrose, 5 mM $Na_3PO_4$, 1 mM NaF, 0.5 mM PMSF, 1 mM $Na_3VO_4$ and protease inhibitors) in a Dounce homogenizer on ice. Homogenates were then centrifuged at 700Xg for 10 min at 4°C to remove nuclei and debris, the supernatants then underwent another centrifugation at 27,200Xg for 90 min at 4°C. Pellets were resuspended in homogenization buffer and protein concentrations were measured. The samples were resolved by SDS-PAGE, transferred to nitrocellulose membrane, and immunoblotted with antibodies. Quantification of immunoblots was done using NIH FIJI/ImageJ. Protein measurements were first normalized to the loading control, NFM, and then as a percentage of the 6-month group (*Figure 1B*) or by setting the mean the of the $Ank1^{F/F}$ control group to 1 (*Figures 2C* and *6F*, and *Figure 6—figure supplement 1C*).

## Plasmids

AnkR-GFP and β1 spectrin-Myc constructs were previously described (*Ho et al., 2014*). The full-length Kv3.1b construct was a gift from Dr. James Trimmer (University of California at Davis). To generate Flag-tagged full-length or truncated Kv3.1b constructs, parts of Kv3.1b were PCR amplified from full-length Kv3.1b and then inserted into p3XFLAG-CMV-7.1 vector. DNA constructs were verified by sequencing (Genewiz).

## Immunoprecipitation

Saline perfused mouse brains were homogenized in 20 mM HEPES pH 7.4, 2 mM EDTA and protease inhibitors in a Dounce homogenizer. 1% (v/v) TX-100 was added to homogenates and solubilized

on a shaker for 30 min at 37 ˚C. Lysates were then centrifuged at 700Xg for 20 min at 4°C to remove nuclei and debris, the supernatants then underwent another centrifugation at 27,200Xg for 60 min at 4°C. Lysates were collected and protein concentrations were measured. The lysates used for immunoprecipitation were prepared by dilution to final protein concentration at 1 mg/ml with lysis buffer (1% (v/v) Triton X-100, 20 mM Tris-HCl pH 8.0, 10 mM EDTA, 150 mM NaCl, 10 mM NaN$_3$ and protease inhibitors). Antibodies (5 ul or 2–4 µg) were added and samples were rotated overnight at 4°C. Protein A (polyclonal antibodies, Thermo Scientific, 20333) or Protein G (monoclonal antibodies, GE Healthcare, 17-0618-01) agarose beads were washed with 1 ml of lysis buffer three times and then rotated with the lysates for 1 hr at 4°C. The beads were then collected and washed with 1 ml of ice-cold lysis buffer seven times and subjected to immunoblotting.

For immunoprecipitation of full-length or truncated Kv3.1b, plasmids for these proteins were co-transfected with AnkR-GFP in HEK293T cells using PEI Max (Polysciences, 24765) according to the manufacturer's instructions. HEK293T cells were obtained from the American Type Culture Collection (ATCC); the identity of the cells was validated by STR profiling and tested negative for mycoplasma contamination. The media was replaced after 16–20 hr of transfection. Cells were lysed at 48 hr of transfection in lysis buffer (50 mM Tris-HCl, pH 8.0, 150 mM NaCl, 0.1% TritonX-100, and 1 mM EDTA with protease inhibitor), and the lysates were centrifuged at 14,000 rpm for 10 min at 4°C. Anti-GFP antibody was mixed with the supernatant and incubated overnight at 4°C. Protein G Mag Sepharose (Cytiva), was first coated with 1 mg/ml of BSA in lysis buffer for 1 hr at 4°C and washed three times with lysis buffer, and then incubated with the mixture of cell lysate and antibody for 1 hr at 4°C. After being washed seven times with lysis buffer, the beads were eluted with 50 µl of 1× Laemmli sample buffer at 95˚C for 5 min. The samples were analyzed by immunoblot using anti-Flag antibody.

## Mass spectrometry

Brain homogenates were prepared as described above before lipid extraction by acetone. Chilled pure acetone was added to homogenate (4:1) then vortexed to precipitate and rotated overnight at 4˚C. Samples were then centrifuged at 27200Xg for 10 min at 4°C and washed twice with chilled acetone and water (4:1). The final pellet was air dried and flash frozen on dry ice. Lysates for AnkR immunoprecipitation were prepared as described above, except the final two washes of the beads were 20 mM TrisHCl pH8, 2 mM CaCl$_2$. The beads were then collected and flash frozen on dry ice.

For digestion, pellets of lipid extracted brain homogenates were resuspended with sonication in 8M guanidinium hydrochloride plus 200 mM ammonium bicarbonate. Protein was reduced by adding 10 mM DTT and incubating at 60°C for 30 min. After that, samples were treated with 20 mM iodoacetamide at room temperature for 30 min and digested with 2% (W/W) Trypsin/LysC mix, mass spectrometry (MS) grade (Promega) for 4 hr at room temperature. Samples were then diluted using 100 mM ammonium bicarbonate so guanidinium hydrochloride concentration was 1M and incubated at 37°C overnight. After this, another 2% W/W aliquot of the digestion enzymes was added, and the digestion was allowed to continue for 4 hr at room temperature. Digested material was recovered using SepPacks C18 cartridges (Waters), eluted in 50% acetonitrile 0.1% formic acid, evaporated and resuspended in 0.1% formic acid for mass spectrometry analysis on a QExactive Plus (Thermo Scientific), connected to a NanoAcquity Ultra Performance UPLC system (Waters). Two µg aliquots of the digests were injected in a 75 µm x 15 cm PepMap RSLC C18 EasySpray column (Thermo Scientific) and peptides resolved in 90 min gradients with 0.1% formic acid in water as mobile phase A and 0.1% formic acid in acetonitrile as mobile phase B. MS was operated in data-dependent mode to automatically switch between MS and MS/MS. The top 10 precursor ions with a charge state of 2 + or higher were fragmented by HCD. A dynamic exclusion window was applied which prevented the same m/z from being selected for 30 s after its acquisition.

For digestion of the immunoprecipitated samples, beads were resuspended in 36 µl 10 mM DTT in 100 mM NH$_4$HCO$_3$ and incubated for 30 min at room temperature. After this, iodoacetamide was added to a final concentration of 15 mM and samples incubated for 30 additional minutes. 0.5 µg of sequencing grade trypsin (Promega) was added to each sample and incubated at 37°C overnight. Supernatants of the beads were recovered, and beads digested again using 0.5 µg trypsin in 100 mM NH$_4$HCO$_3$ for 2 hr. Peptides from both consecutive digestions were recovered by solid phase extraction using C18 ZipTips (Millipore), eluted in 2x7 µl aliquots of 50% MeCN 0.1% formic acid, dried and resuspended in 2.5 µl 0.1% formic acid for mass spectrometry analysis. Peptides were

separated using a 75 µm x 50 cm PepMap RSLC C18 EasySpray column (Thermo Scientific) using 3 hr gradients with 0.1% formic acid in water as mobile phase A and 0.1% formic acid in acetonitrile as mobile phase B, for analysis in a Orbitrap Lumos Fusion (Thermo Scientific) in positive ion mode. MS was operated in 3 s cycles in data-dependent mode to automatically switch between MS and MS/MS, with a charge state of 2+ or higher were fragmented by HCD. A dynamic exclusion window was applied which prevented the same m/z from being selected for 30 s after its acquisition.

In both cases, peak lists were generated using PAVA in-house software (*Guan et al., 2011*). Generated peak lists were searched against the *Mus musculus* subset of the UniprotKB database (UniProtKB.2013.6.17 for the full brain samples and UniProtKB.2017.11.01 for Ip samples), using Protein Prospector (*Clauser et al., 1999*) with the following parameters: Enzyme specificity was set as Trypsin, and up to two missed cleavages per peptide were allowed. Carbamidomethylation of cysteine residues was allowed as fixed modification. N-acetylation of the N-terminus of the protein, loss of protein N-terminal methionine, pyroglutamate formation from peptide N-terminal glutamines, and oxidation of methionine were allowed as variable modifications. Mass tolerance was 10 ppm in MS and 30 ppm in MS/MS. The false positive rate was estimated by searching the data using a concatenated database which contains the original UniProtKB database, as well as a version of each original entry where the sequence has been randomized. A 1% FDR was permitted at the protein and peptide level.

## Brain slice electrophysiology

All electrophysiological experiments were performed and analyzed blind to the genotypes. Mice were anesthetized by an intraperitoneal injection of a ketamine and xylazine mix (80 mg/kg and 16 mg/kg, respectively) and transcardially perfused with cold (0–4°C) slice cutting solution containing 80 mM NaCl, 2.5 mM KCl, 1.3 mM $NaH_2PO_4$, 26 mM $NaHCO_3$, 4 mM $MgCl_2$, 0.5 mM $CaCl_2$, 20 mM D-glucose, 75 mM sucrose and 0.5 mM sodium ascorbate (315 mosmol, pH 7.4, saturated with 95% $O_2$/5% $CO_2$). Brains were removed and sectioned in the cutting solution with a VT1200S vibratome (Leica) to obtain 300 µm coronal slices. Slices containing primary somatosensory cortex were collected and incubated in a custom-made interface holding chamber saturated with 95% $O_2$/5% $CO_2$ at 34°C for 30 min and then at room temperature for 20 min to 6 hr until they were transferred to the recording chamber. Prior to moving slices to the recording chamber, 500 µl of fluorescein labeled WFA solution (200 mg/ml, in oxygenated cutting solution) was dropped on top of slices in holding chamber and incubated for 30–45 min to label perineuronal nets. After the incubation period slices were rinsed three to four times with oxygenated cutting solution before transferring to the recording chamber.

Recordings were performed at 32°C on submerged slices in artificial cerebrospinal fluid (ACSF) containing 119 mM NaCl, 2.5 mM KCl, 1.3 mM $NaH_2PO_4$, 26 mM $NaHCO_3$, 1.3 mM $MgCl_2$, 2.5 mM $CaCl_2$, 20 mM D-glucose and 0.5 mM sodium ascorbate (305 mosmol, pH 7.4, saturated with 95% $O_2$/5% $CO_2$, perfused at 3 ml/min). For whole-cell recordings, a $K^+$-based pipette solution containing 142 mM $K^+$-gluconate, 10 mM HEPES, 1 mM EGTA, 2.5 mM $MgCl_2$, 4 mM ATP-Mg, 0.3 mM GTP-Na, 10 mM $Na_2$-phosphoCreatine (295 mosmol, pH 7.35) or a $Cs^+$-based pipette solution containing 121 mM $Cs^+$-methanesulfonate, 10 mM HEPES, 10 mM EGTA, 1.5 mM $MgCl_2$, 4 mM ATP-Mg, 0.3 mM GTP-Na, 10 mM $Na_2$-phosphoCreatine, and 2 mM QX314-Cl (295 mosmol, pH 7.35) was used. Membrane potentials were not corrected for liquid junction potential (experimentally measured as 12.5 mV for the $K^+$-based pipette solution and 9.5 mV for the $Cs^+$-based pipette solution).

Neurons were visualized with video-assisted infrared differential interference contrast imaging and WFA+ neurons were identified by epifluorescence imaging under a water immersion objective (40x, 0.8 numerical aperture) on an upright SliceScope Pro 1000 microscope (Scientifica) with an infrared IR-1000 CCD camera (DAGE-MTI). Data were acquired at 10 kHz and low-pass filtered at 4 kHz with an Axon Multiclamp 700B amplifier and an Axon Digidata 1440 Data Acquisition System under the control of Clampex 10.7 (Molecular Devices). Fast-spiking cells were defined based on their response to a 500 ms supra-threshold current injection. These cells display a fast action potential along with a very high maximal firing frequency and minimal frequency or amplitude adaptation (*McCormick et al., 1985*).

Neuronal intrinsic excitability was examined with the $K^+$-based pipette solution. The resting membrane potential was recorded in the whole-cell current clamp mode within the first minute after break-in. After balancing the bridge, the input resistance and membrane capacitance were

measured by injecting a 500 ms hyperpolarizing current pulse (50–100 pA) to generate a small membrane potential hyperpolarization (2–10 mV) from the resting membrane potential. Depolarizing currents were increased in 5- or 10 pA steps to identify rheobase currents. To generate the current-firing frequency curves, the resting membrane potential of neurons was held at –75 mV and 500 ms depolarizing current pulses were increased by 50 pA steps from 0 to 1450 pA. If depolarization block occurred prior to 1450 pA, then recording was stopped.

To record miniature synaptic currents, whole-cell voltage clamp recordings of WFA+ cells were performed with the $Cs^+$-based pipette solution in ACSF containing 1 µM tetrodotoxin. mEPSCs and mIPSCs were recorded for 2–3 min at the reversal potential for inhibition (–70 mV) and excitation (+10 mV), respectively.

## Electrophysiology data analysis

Data were analyzed offline by AxoGraph X (AxoGraph Scientific). The single action potential generated by the rheobase current was used to analyze action potential characteristics. Action potential threshold was defined as the voltage at which the first derivative of voltage over time exceeded 20 V/s. The action potential latency was determined as the time between the onset of current injection and action potential threshold. Action potential amplitude was determined as the voltage difference between the action potential threshold and peak. Action potential half-width was measured as the duration of the action potential at the voltage halfway between the action potential threshold and peak. Afterhyperpolarization (AHP) amplitude was determined as the minimum voltage following the action potential peak subtracted from the action potential threshold. AHP time was determined as the time between action potential threshold and the negative peak of AHP.

To analyze spike trains, action potentials were detected using Axograph X event detection with a fixed amplitude template defined by the shape of the action potential. The current-firing frequency curves were generated by measuring the frequency of action potentials for each current injection. Spike frequency adaptation and amplitude adaptation were determined from the spike train evoked by the currents that are two times of the action potential threshold currents. Spike frequency adaptation was measured by determining the percent decrease between the inverse of the average of the first five inter-spike intervals and the inverse of the average of the last five inter-spike intervals of the 500 ms pulse. Amplitude adaptation was determined by baselining the trace at the action potential threshold for the first action potential and then calculating the percent decrease between the amplitude of the first action potential and the last one. The firing frequency within the first 100 ms of current injection was also determined. Depolarization block current was determined as the minimal current that caused the cell to reach depolarization block during the 500 ms current pulse. If depolarization block was not reached by 1450 pA current injection, 1450 pA was recorded as the value.

To detect miniature synaptic events, data were digitally low-pass filtered at 2 kHz offline and events were detected by a scaled-template algorithm (AxoGraph X). The parameters of the template for mEPSCs are: length, 5 ms; baseline, 1.5 ms; amplitude, –2 pA; rise time, 0.2 ms; and decay time, 1 ms with a detection threshold of –3.25 s.D. The parameters of the template for mIPSCs are: length, 10 ms; baseline, 3 ms; amplitude, 2 pA; rise time, 0.27 ms; and decay time, 3.7 ms with a detection threshold of 3. The integrated charge per unit time for mEPSC or mIPSC was determined by multiplying the frequency of mEPSC or mIPSC by the average charge of mEPSCs or mIPSCs, respectively.

## Statistical analyses

No statistical methods were used to pre-determine sample sizes, but our sample sizes are similar to those previously reported (*Chiang et al., 2018*; *Ho et al., 2014*). Sets of age-matched conditional knockout mice and their controls were randomly collected from the same litter or from two litters that had close dates of birth. Data were collected and processed randomly and were analyzed using Microsoft Excel and GraphPad Prism. Except for perineuronal net quantifications, researchers were not 'blinded' to the conditions of the experiments for data collection and analysis. Unless otherwise stated, unpaired *t*-tests with an alpha of 0.05 were used for statistical analysis, data distributions were not assumed to be normal. Multiple comparisons were corrected using the Holm-Šídák method. All error bars are SEM unless otherwise indicated. p Values and adjusted p values found

significant are indicated in figures for post hoc comparisons. Full statistical analyses and test parameters are provided in the extended data.

## Acknowledgements

The work reported here was supported by the following research grants from the National institutes of Health: R01 NS044916 (MNR), R01 GM103481 (ALB), R01 MH117089 (MX), R01 NS100893 (MX), F31 NS100300 (SRS), F30 MH118804 (CML), RF1 NS118584 (MDC). This work was also supported by the Dr. Miriam and Sheldon G Adelson Medical Research Foundation (ALB and MNR), the HMH Clinician Scientist Award (MDC), and a Houston Methodist Hospital/Baylor College of Medicine collaborative pilot grant in Alzheimer's disease and related dementias. CML is part of the Baylor College of Medicine Medical Scientist Training Program and McNair MD/PhD Student Scholars supported by the McNair Medical Institute at the Robert and Janice McNair Foundation. MX is a Caroline DeLuca Scholar.

## Additional information

### Funding

| Funder | Grant reference number | Author |
| --- | --- | --- |
| National Institute of Neurological Disorders and Stroke | NS044916 | Matthew N Rasband |
| National Institute of General Medical Sciences | GM103481 | Alma L Burlingame |
| National Institute of Mental Health | MH117089 | Mingshan Xue |
| National Institute of Neurological Disorders and Stroke | NS100893 | Mingshan Xue |
| National Institute of Neurological Disorders and Stroke | NS100300 | Sharon Stevens |
| National Institute of Mental Health | MH118804 | Colleen M Longley |
| National Institute of Neurological Disorders and Stroke | NS118584 | Matthew Cykowski |
| Dr. Miriam and Sheldon G. Adelson Medical Research Foundation | | Alma L Burlingame Matthew N Rasband |

The funders had no role in study design, data collection and interpretation, or the decision to submit the work for publication.

### Author contributions

Sharon R Stevens, Conceptualization, Data curation, Formal analysis, Validation, Investigation, Visualization, Methodology, Writing - original draft, Writing - review and editing; Colleen M Longley, Formal analysis, Validation, Investigation, Methodology, Writing - original draft, Writing - review and editing; Yuki Ogawa, Formal analysis, Validation, Investigation, Methodology, Writing - review and editing; Lindsay H Teliska, Supna Nair, Investigation; Anithachristy S Arumanayagam, Validation, Investigation; Juan A Oses-Prieto, Resources, Data curation, Investigation, Writing - review and editing; Alma L Burlingame, Resources, Supervision, Funding acquisition, Writing - review and editing; Matthew D Cykowski, Mingshan Xue, Supervision, Funding acquisition, Investigation, Visualization, Methodology, Writing - review and editing; Matthew N Rasband, Conceptualization, Data curation, Supervision, Funding acquisition, Methodology, Writing - original draft, Project administration, Writing - review and editing

## Author ORCIDs

Sharon R Stevens (ID) https://orcid.org/0000-0003-2238-8029
Colleen M Longley (ID) http://orcid.org/0000-0001-8326-6143
Lindsay H Teliska (ID) https://orcid.org/0000-0003-1733-6910
Supna Nair (ID) http://orcid.org/0000-0001-6499-0099
Juan A Oses-Prieto (ID) http://orcid.org/0000-0003-4759-2341
Mingshan Xue (ID) http://orcid.org/0000-0003-1463-8884
Matthew N Rasband (ID) https://orcid.org/0000-0001-8184-2477

## Ethics

Animal experimentation: All experiments were conducted in compliance with the National Institutes of Health Guide for the Care and Use of Laboratory Animals and were approved by the Animal Care and Use Committee at Baylor College of Medicine under approval AN4634.

## Decision letter and Author response

Decision letter https://doi.org/10.7554/eLife.66491.sa1
Author response https://doi.org/10.7554/eLife.66491.sa2

## Additional files

### Supplementary files

• Supplementary file 1. Intrinsic properties of WFA$^+$ cells in *Ank1$^{F/F}$*, *Ank1$^{+/+}$;Dlx5/6-Cre*, and *Ank1$^{F/F}$;Dlx5/6-Cre* mice. Data are from 3 *Ank1$^{F/F}$*, 2 *Ank1$^{+/+}$;Dlx5/6-Cre*, and 4 *Ank1$^{F/F}$;Dlx5/6-Cre* mice, and are reported as mean ± SEM (number of cells). Bolded p values indicate significance.

• Supplementary file 2. Abbreviations used throughout the paper.

• Transparent reporting form

## Data availability

All data generated or analysed during this study are included in the manuscript and supporting files. Source data files have been provided for all Figures.

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

# Appendix 1

**Appendix 1—key resources table**

| Reagent type (species) or resource | Designation | Source or reference | Identifiers | Additional information |
|---|---|---|---|---|
| gene (*human*) | *ANK1* | NCBI.NLM.NIH.gov/gene | Gene ID: 286 HGNC: HGNC:492 | |
| gene (*Mus musculus*) | *Ank1* | NCBI.NLM.NIH.gov/gene | Gene ID: 11733 MGI: MGI:88024 | |
| Strain, strain background (*Mus musculus*, male and female) | C57BL/6J, Wild type, WT | The Jackson Laboratory | JAX:000664 | See Materials and methods, Section Animals |
| Strain, strain background (*Mus musculus*, female) | *Nestin-Cre, Nes-Cre* | The Jackson Laboratory | JAX:003771 | See Materials and methods, Section Animals |
| Strain, strain background (*Mus musculus*, male) | *Dlx5/6-Cre* | The Jackson Laboratory | JAX:008199 | See Materials and methods, Section Animals |
| Strain, strain background (*Mus musculus*, male) | *Chat-Cre ChAT-Cre* | The Jackson Laboratory | JAX:006410 | See Materials and methods, Section Animals |
| Strain, strain background (*Mus musculus*, male and female) | *Ank1$^{pale/pale}$ Ank1-KO* | The Jackson Laboratory | JAX:009157 | See Materials and methods, Section Animals |
| Strain, strain background (*Mus musculus*, male and female) | *Ank1$^{F/F}$* | This paper | JAX:036512 | See Materials and methods, Section Animals Dr. Matthew Rasband (Baylor College of Medicine) |
| Strain, strain background (*Mus musculus*, male and female) | *Ank3$^{F/F}$* | The Jackson Laboratory | JAX:029797 | See Materials and methods, Section Animals |
| Sequence-based reagent | Genotyping primer for *Ank1$^{F/F}$* mouse (sense) | This paper | | See Materials and methods, Section Animals 5'-GGGAAAC TCCACAGAGCCTGACGGG TCAGT-3' |
| Sequence-based reagent | Genotyping primer for *Ank1$^{F/F}$* mouse (missense) | This paper | | See Materials and methods, Section Animals 5'- GGC GTC CCTATGTTC CATCCTATA GATGACT-3' |
| Transfected construct (*M. musculus*) | Full-length AnkR-GFP | *Ho et al., 2014* | | See Materials and methods, Section Plasmids |
| Transfected construct (*M. musculus*) | β1 spectrin-Myc | *Ho et al., 2014* | | See Materials and methods, Section Plasmids |
| Transfected construct (*M. musculus*) | Full-length Kv3.1b | Dr. James Trimmer (University of California at Davis) | | See Materials and methods, Section Plasmids |
| Transfected construct (*M. musculus*) | Full-length Kv3.1b-Flag | This paper | | See Materials and methods, Section Plasmids p3XFLAG-CMV-7.1 vector |
| Transfected construct (*M. musculus*) | Truncated Kv3.1b constructs | This paper | | See Materials and methods, Section Plasmids p3XFLAG-CMV-7.1 vector |
| cell line (*Homo-sapiens*) | HEK293T | ATCC | | See Materials and methods, Section Immunoprecipitation |

*Continued on next page*

*Appendix 1—key resources table continued*

| Reagent type (species) or resource | Designation | Source or reference | Identifiers | Additional information |
|---|---|---|---|---|
| Antibody | Ankyrin-R AnkR (mouse monoclonal) | UC Davis/NIH NeuroMab Facility Cat# 75–380 | RRID:AB_2491109 | IF (1:250) IB (1:500) IP (5 ul) |
| Antibody | β1 spectrin (mouse monoclonal) | UC Davis/NIH NeuroMab Facility Cat# 73–374 | RRID:AB_2315814 | IF (1:250) IB (1:1000) IP (5 ul) |
| Antibody | Ankyrin-G AnkG (mouse monoclonal) | UC Davis/NIH NeuroMab Facility Cat# 73–146 | RRID:AB_10697718 | IF (1:500) |
| Antibody | Parvalbumin Pv (mouse monoclonal) | UC Davis/NIH NeuroMab Facility Cat# 73–455 | RRID:AB_2629420 | IF (1:250) |
| Antibody | Actin (mouse monoclonal) | Millipore Cat# MAB1501 | RRID:AB_2223041 | IB (1:1000) |
| Antibody | Tenascin-R TnR (mouse monoclonal) | R and D Systems Cat# MAB1624 | RRID:AB_2207001 | IF (1:500) |
| Antibody | Aggrecan ACAN Acan (mouse monoclonal) | Millipore Cat# AB1031 | RRID:AB_90460 | IF (1:500) |
| Antibody | Brevican BCAN Bcan (mouse monoclonal) | UC Davis/NIH NeuroMab Facility Cat# 75–294 | RRID:AB_2315824 | IF (1:500) |
| Antibody | NrCAM (mouse monoclonal) | R and D Systems Cat# MAB2034 | RRID:AB_2267411 | IF (1:500) IB (1:1000) IP (5 ul) |
| Antibody | Kv3.1b (mouse monoclonal) | UC Davis/NIH NeuroMab Facility Cat# N16B/8 | RRID:AB_2750730 | IF (1:500) IB (1:1000) IP (5 ul) |
| Antibody | Kv3.2 (mouse monoclonal) | UC Davis/NIH NeuroMab Facility Cat# N410/17 | RRID:AB_2877619 | IF (1:250) |
| Antibody | Kv3.3 (mouse monoclonal) | Antibodies-Online Cat# ABIN572016 | RRID:AB_10782137 | IF (1:500) |
| Antibody | Kv7.2 (mouse monoclonal) | UC Davis/NIH NeuroMab Facility Cat# N26A/23 | RRID:AB_2750761 | IF (1:500) |
| Antibody | Pan sodium channel PanNav (mouse monoclonal) | UC Davis/NIH NeuroMab Facility Cat# N419/78 | RRID:AB_2877588 | IF (1:250) |
| Antibody | Flag-tag or DDDDK-tag (mouse monoclonal) | MBL International Cat# M185-3L | RRID:AB_11123930 | IB (1:1000) |
| Antibody | Ankyrin-R AnkR (rabbit polyclonal) | *Ho et al., 2014* | RRID:AB_2833096 | IF (1:500) IB (1:1000) IP (5 ul) |
| Antibody | Ankyrin-R, AnkR, Ank1 (rabbit polyclonal) | Thermo Fisher Scientific Cat# PA5-63372 | RRID:AB_2638015 | IF (1:500) IB (1:1000) |
| Antibody | Neurofilament M, NFM (rabbit polyclonal) | Millipore Cat# AB1987 | RRID:AB_91201 | IB (1:2000) |

*Continued on next page*

*Appendix 1—key resources table continued*

| Reagent type (species) or resource | Designation | Source or reference | Identifiers | Additional information |
|---|---|---|---|---|
| Antibody | Somatostatin, SST (rabbit polyclonal) | Peninsula Laboratories Cat# T-4103.0050 | RRID:AB_ 518614 | IF (1:3000) |
| Antibody | Parvalbumin Pv (rabbit polyclonal) | Novus Cat# NB120-11427 | RRID:AB_ 791498 | IF (1:500) |
| Antibody | Versican, VCAN (rabbit polyclonal) | Millipore Cat# AB1032 | RRID:AB_ 11213831 | IF (1:500) |
| Antibody | PlexinA4, PlxnA4 (rabbit polyclonal) | Abcam Cat# ab39350 | RRID:AB_ 944890 | IF (1:500) IB (1:1000) IP (5 ul) |
| Antibody | Neruropilin-1, Nrp1 (rabbit polyclonal) | GeneTex Cat# GTX16786 | RRID:AB_ 422398 | IF (1:500) IB (1:1000) IP (5 ul) |
| Antibody | Kv3.1 (rabbit polyclonal) | LSBio (LifeSpan) Cat#LS-C322374-200 | RRID:AB_ 2891125 | IF (1:100) |
| Antibody | Kv3.1b (rabbit polyclonal) | Alomone Labs Cat# APC-014 | RRID:AB_ 2040166 | IF (1:500) IB (1:1000) IP (5 ul) |
| Antibody | Kv3.3 (rabbit polyclonal) | Alomone Labs Cat# APC-102 | RRID:AB_ 2040170 | IF (1:500) |
| Antibody | GFP (rabbit polyclonal) | Thermo Fisher Scientific, Cat# A-11122 | RRID:AB_ 221569 | IB (1:1000) |
| Antibody | Neurofascin, NF, NF186 (chicken polyclonal) | R and D Systems Cat# AF3235 | RRID:AB_ 10890736 | IF (1:500) |
| Other | Fluorescein WFA, *Wisteria floribunda agglutinin* | Vector Laboratories Cat# FL-1351 | RRID:AB_ 2336875 | Lectin IF (1:250) |
| Other | Fluorescein WFA, *Wisteria floribunda agglutinin* | Bioworld Cat# 21761065–1 | RRID:AB_ 2833087 | Lectin IF (1:100) |
| Other | Texas-Red WFA, *Wisteria floribunda agglutinin* | EY Laboratories Cat# F-3101–1 | RRID:AB_ 2315605 | Lectin IF (1:250) |
| Software, algorithm | Fiji, Image J | NIH | RRID:SCR_ 002285 | |
| Software, algorithm | Prism | Graph Pad | RRID:SCR_ 011323 | |

IF: Immunofluorescence; IB: Immunoblot; IP:Immunoprecipitation

