## [Decision Letter]

**Acceptance summary:**

This study describes the role of Ankyrin-R, a linker protein that joins integral membrane proteins to the cytoskeletal network, in parvalbumin (PV) interneurons. Using cell specific knockouts, physiology, histology and behavior, the authors provide evidence that Ankryin-R localizes Kv3.1 channels at axon initial segments and is necessary for proper formation of perineuronal nets, indicating that it serves a critical role in controlling the structure and function of this class of interneurons.

**Decision letter after peer review:**

Thank you for submitting your article "Ankyrin-R regulates fast-spiking interneuron excitability through perineuronal nets and Kv3.1b K + channels" for consideration by *eLife*. Your article has been reviewed by 2 peer reviewers, and the evaluation has been overseen by a Reviewing Editor and Richard Aldrich as the Senior Editor. The reviewers have opted to remain anonymous.

Essential Revisions:

1) Substantial revisions in data analysis, statistical comparisons and description of methods are required.

a. Description of methods for statistical analysis is either unclear or incorrect statistical analyses are being performed for some experiments. For example, the use of both Cre only and floxed only controls for the mouse studies is rigorous and is a major strength. However, in some cases only select statistical comparisons are being shown when it is clear that the null mouse is only different from one of the two controls. For example, the null line is being compared to floxed controls in Figure 3G WFA+/PV+, but to the Cre control for WFA+, PV-. It is indicated in the text that these comparisons are made using t test when three groups are being compared (flox only, cre only, and Ank1 null). The description of what tests are being used for each panel should be included and the appropriate tests should be used to compare groups (i.e. ANOVA or Kruskal-Wallis, depending on data distribution). This applies to Figures 3, S3, 4, S4, 5, S5, and table S1. In addition, Line 815 states: "Data distributions were assumed to be normal", but clearly the validity of this assumption has to be tested.

b. In several figures, ankyrin-R is being used as a marker for this subset of neurons and is being compared to the Ank1 null mouse. However, once ankyrin-R is deleted, it is unclear which cells are being analyzed, since no other marker is being used for this cell population (i.e. PV, WFA, etc.). For example, Figures S1, 6, S6.

c. The use of negative controls for immunoprecipitation experiments is a strength, however, generally these negative control IP protein is not being shown. This makes it difficult to interpret whether this is a suitable control. For example, was Caspr efficiently IPed in Figure 2B or 2K. This is especially important because the apparent strength of some of the interactions seems relatively low. Similarly, it is unclear why ankyrin-R is running at a different molecular weigh in the PlexinA4 IP (Figure 2L).

d. Consistent quantification of the microscopy images would significantly strengthen conclusions. For example, the authors claim that somatodendritic NrCAM, but not that at the AIS, was decreased in Ank1-deficient mice (Figure 2J), but no quantification is provided (nor is another marker for the normally ankyrin-R-positive cells, see point 2). No description is provided for how it was determined if a cell was positive for markers like PV or WFA. Additional information should be provided to explain how images were thresholded and counted. The conclusion that the perineuronal net is more fragmented after ankyrin R deletion is unconvincing. High-resolution imaging followed by three dimensional reconstruction seems necessary. Ideally, super-resolution imaging should be performed or these illustration and quantification of these data extended to support this claim. The subcellular distribution of ankyrin R in PV+ GABAergic interneurons is insufficiently analyzed. If the authors want to make the point that ankyrin is located "somatodendritically", they need to quantify the expression in dendrites.

e. The results shown here implicate ankyrin-R in the normal protein levels of Kv3.1b, with null mice showing ~50% decrease in channel levels. It is surprising that significant changes in potassium channel levels would not have an impact on resting membrane potential or input resistance. It would be useful to see ensemble potassium currents from these cells along with phase plane plots of the action potential. Similarly, it is surprising that loss of 50% of Kv3.1b does not have an impact on PV neuron fast-spiking, given the critical role of this channel in repolarization. It would be helpful if the authors could address whether there is compensation from Kv3.1a. 3. Ankyrin R may affect other channels, such as sodium channels, to alter the firing behavior of PV interneurons. This important point is unaddressed in the present paper. Second, Kv 3.2 may play a role in regulation of excitability, independently of the reported effects. Kv3.2 Is functionally redundant with Kv 3.1, and is coexpressed with Kv3.1 in several types of GABAergic interneurons. Clearly, the expression of these two major players needs to be addressed. It would be helpful if the authors could discussion what aspects of channel function are altered by interaction with ankryin-R, such as current density, channel gating, or both are changed.

f. The conclusion that input synapses of PV+ interneurons are unaffected by ankyrin R is on weak grounds, and is partially contradictory with previous literature. Ideally, the authors should perform paired recording experiments to address this issue or temper this claim.

g. The authors claim that 70% of ankyrin R-positive neurons are immunopositive for parvalbumin. The identity of the remaining 30% is unclear. From my point of view, it would be important to fill this gap in the authors' conclusion. Immunolabeling for somatostatin could provide a quick answer.

2) The Introduction and Discussion should be revised.

a) The presentation of manuscript needs to be improved. The Introduction would benefit form a more systematic explanation of the structure and function of different ankyrin genes.

b) The Discussion is too focused on disease aspects, which are only tangentially related to the contents of the Results.

c) The readability would be improved by removing some acronyms.

d) The list of references is excessive.

*Reviewer #1:*

Stevens et al., examined the function of ankyrin-R, product of the Ank1 gene, in normal function of a class of critical inhibitory neurons that express the marker protein, parvalbumin (PV). The athors have shown that ankyrin-R is highly expressed in PV neurons where it binds to and controls localization of a specific voltage-gated potassium channel. In addition, they show that deletion of Ank1 causes several deficits in PV neuron function and structure, especially in their associated extracellular matrix-derived structures called perineuronal nets. Both PV neurons and perineuronal nets are linked to a number of neuropsychiatric and developmental disorders, increasing the impact of these findings. This work represents one of the first descriptions of a normal physiological role for ankyrin-R in the nervous system and expands our knowledge of the ever-growing list of neuronal functions for the ankyrin family. In general, the experiments were rigorously controlled and the quality of the data is strong. However, several major weaknesses below significantly weaken the conclusions of this manuscript.

1) Description of methods for statistical analysis is either unclear or incorrect statistical analyses are being performed for some experiments. For example, the use of both Cre only and floxed only controls for the mouse studies is rigorous and is a major strength. However, in some cases only select statistical comparisons are being shown when it is clear that the null mouse is only different from one of the two controls. For example, the null line is being compared to floxed controls in Figure 3G WFA+/PV+, but to the Cre control for WFA+, PV-. It is indicated in the text that these comparisons are made using t test when three groups are being compared (flox only, cre only, and Ank1 null). The description of what tests are being used for each panel should be included and the appropriate tests should be used to compare groups (i.e. ANOVA or Kruskal-Wallis, depending on data distribution). This applies to Figures 3, S3, 4, S4, 5, S5, and table S1).

2) In several figures, ankyrin-R is being used as a marker for this subset of neurons and is being compared to the Ank1 null mouse. However, once ankyrin-R is deleted, it is unclear which cells are being analyzed, since no other marker is being used for this cell population (i.e. PV, WFA, etc.). For example, Figures S1, 6, S6.

3) The use of negative controls for immunoprecipitation experiments is a strength, however, generally these negative control IP protein is not being shown. This makes it difficult to interpret whether this is a suitable control. For example, was Caspr efficiently IPed in Figure 2B or 2K. This is especially important because the apparent strength of some of the interactions seems relatively low. Similarly, it is unclear why ankyrin-R is running at a different molecular weigh in the PlexinA4 IP (Figure 2L).

4) Consistent quantification of the microscopy images would significantly strengthen conclusions. For example, the authors claim that somatodendritic NrCAM, but not that at the AIS, was decreased in Ank1-deficient mice (Figure 2J), but no quantification is provided (nor is another marker for the normally ankyrin-R-positive cells, see point 2). No description is provided for how it was determined if a cell was positive for markers like PV or WFA. It is unclear how were the images thresholded and counted.

5) The results shown here implicate ankyrin-R in the normal protein levels of Kv3.1b, with null mice showing ~50% decrease in channel levels. It is surprising that significant changes in potassium channel levels would not have an impact on resting membrane potential or input resistance. It would be useful to see ensemble potassium currents from these cells along with phase plane plots of the action potential. Similarly, it is surprising that loss of 50% of Kv3.1b does not have an impact on PV neuron fast-spiking, given the critical role of this channel in repolarization. It is unclear whether there is compensation from Kv3.1a.

*Reviewer #2:*

The paper by Stevens et al. examines the role of ankyrin R in parvalbumin-expressing, fast-spiking GABAergic interneurons. To address this question, the authors use immunocytochemistry, proteomics, and electrophysiology. The main findings are:

– Deletion of ankyrin R disrupts perineuronal nets, decreases anxiety-like behaviors, and changes the intrinsic excitability of PV+ interneurons.

– Deletion of ankyrin R leads to a significant reduction of Kv3.1 potassium channel density.

– Interaction between ankyrin R and Kv3.1 occurs via a novel ankyrin binding motif.

Based on these results, the authors conclude that ankyrin R regulates function of PV+ interneurons by organizing cell adhesion molecules, perineuronal nets, and ion channels. Overall, this is a potentially interesting paper. Although the role of perineuronal nets and Kv3 channels in the excitability and fast spiking of PV + interneurons is well established, in the finding that ankyrin R regulates these properties seems novel.

1. The conclusion that input synapses of PV+ interneurons are unaffected by ankyrin R is on weak grounds, and is partially contradictory with previous literature. Ideally, the authors should perform paired recording experiments to address this issue.

2. The conclusion that the perineuronal net is more fragmented after ankyrin R deletion is unconvincing. High-resolution imaging followed by three dimensional reconstruction seems necessary. Ideally, super-resolution imaging should be performed.

3. The authors describe several links between ankyrin R and Kv3.1. However, other channels may be affected as well. First, ankyrin R may affect sodium channel density in interneurons. This important point is unaddressed in the present paper. Second, Kv 3.2 may play a role in regulation of excitability, independently of the reported effects. Kv3.2 Is functionally redundant with Kv 3.1, and is co-expressed with Kv3.1 in several types of GABAergic interneurons. Clearly, the expression of these two major players would need to be addressed.

4. The authors claim that 70% of ankyrin R-positive neurons are immunopositive for parvalbumin. The identity of the remaining 30% is unclear. From my point of view, it would be important to fill this gap in the authors' conclusion. Immunolabeling for somatostatin could provide a quick answer.

5. The subcellular distribution of ankyrin R in PV+ GABAergic interneurons is insufficiently analyzed. If the authors want to make the point that ankyrin is located "somatodendritically", they would need to quantify the expression in dendrites.

6. Functional conclusions stand and fall with the analysis of the current-clamp data. Conclusions may be corroborated by voltage-clamp analysis of Kv3 currents, preferentially under optimal voltage-clamp conditions in nucleated patches. Such experiments may reveal whether current density, channel gating, or both are changed.

7. There are some concerns about statistics. Line 815 states: "Data distributions were assumed to be normal" , but clearly the validity of this assumption has to be tested. Furthermore, in line 1198, the authors state that an ANOVA or a (nonparametric) Kruskal-Wallis test was used. These discrepancies should be resolved. Finally, it would need to be stated more clearly if and how correction for multiple comparisons was done in the pairwise tests.

---

## [Author Response]

Reviewer #1:Stevens et al., examined the function of ankyrin-R, product of the Ank1 gene, in normal function of a class of critical inhibitory neurons that express the marker protein, parvalbumin (PV). The athors have shown that ankyrin-R is highly expressed in PV neurons where it binds to and controls localization of a specific voltage-gated potassium channel. In addition, they show that deletion of Ank1 causes several deficits in PV neuron function and structure, especially in their associated extracellular matrix-derived structures called perineuronal nets. Both PV neurons and perineuronal nets are linked to a number of neuropsychiatric and developmental disorders, increasing the impact of these findings. This work represents one of the first descriptions of a normal physiological role for ankyrin-R in the nervous system and expands our knowledge of the ever-growing list of neuronal functions for the ankyrin family. In general, the experiments were rigorously controlled and the quality of the data is strong. However, several major weaknesses below significantly weaken the conclusions of this manuscript.1) Description of methods for statistical analysis is either unclear or incorrect statistical analyses are being performed for some experiments. For example, the use of both Cre only and floxed only controls for the mouse studies is rigorous and is a major strength. However, in some cases only select statistical comparisons are being shown when it is clear that the null mouse is only different from one of the two controls. For example, the null line is being compared to floxed controls in Figure 3G WFA+/PV+, but to the Cre control for WFA+, PV-. It is indicated in the text that these comparisons are made using t test when three groups are being compared (flox only, cre only, and Ank1 null). The description of what tests are being used for each panel should be included and the appropriate tests should be used to compare groups (i.e. ANOVA or Kruskal-Wallis, depending on data distribution). This applies to Figures 3, S3, 4, S4, 5, S5, and table S1).

We apologize for the confusion. Although comparisons were made for all genotypes, we only showed (on the figures) those that were significant. The results of all comparisons are included in the extended data file. Nevertheless, the reviewer is correct that in some instances we did not perform multiple comparisons when multiple genotypes were analyzed. We have fixed this. We repeated all statistical comparisons and performed multiple unpaired t-tests with the Holm-Sidak method for multiple comparisons. In the case of electrophysiology, we tested for normality on each data set and performed either Ordinary One-Way ANOVA or Kruskal-Wallis based on the results. Again, on the figures we show only those comparisons that were significantly different. All statistical comparisons for all results are included in the extended data file.

2) In several figures, ankyrin-R is being used as a marker for this subset of neurons and is being compared to the Ank1 null mouse. However, once ankyrin-R is deleted, it is unclear which cells are being analyzed, since no other marker is being used for this cell population (i.e. PV, WFA, etc.). For example, Figures S1, 6, S6.

The magnifications of all images shown include many interneurons. In all panels we show AnkR neuron Figures labeling is completely gone. Nevertheless, we now provide double-labeled images in a new Figure 3 – Supplement 1 that includes WFA, confirming the loss of AnkR from WFA+ cells in cKO mice. In 6 and S6 (now Figure 6 and Figure 6 – Supplement 1) quantifications are not made on a cell-by-cell basis, but rather an entire field of view after thresholding of the image based on the control brain sections. Expanded explanation and details are included in the revised methods.

3) The use of negative controls for immunoprecipitation experiments is a strength, however, generally these negative control IP protein is not being shown. This makes it difficult to interpret whether this is a suitable control. For example, was Caspr efficiently IPed in Figure 2B or 2K. This is especially important because the apparent strength of some of the interactions seems relatively low. Similarly, it is unclear why ankyrin-R is running at a different molecular weigh in the PlexinA4 IP (Figure 2L).

Unfortunately, we did not perform immunoblot for Caspr and do not have these data. Nevertheless, we now include the IgG band for the controls. We replaced the label as IgG IP for the control. Since the concern is nonspecific interaction, any IgG can serve as a control (frequently pre-immune serum is used as a control). The IgG bands are now shown to demonstrate immunoprecipitation of the primary antibodies. We agree with the reviewer that the interactions shown by IP are quite weak. This is not surprising. Co-IPs with ankyrin scaffolding proteins are notoriously difficult. For example, co-IPs between known ankyrin binding partners (e.g. Na^+^ channels, KCNQ2/3 K^+^ channels, and L1 family cell adhesion molecules like NF186 and NrCAM) are extremely difficult. Previous studies have used surface recruitment assays in heterologous cells, purified protein fragments, or genetic interactions to establish their binding (see for example Pan et al., J Neurosci 2006; Zhang et al., J Biol Chem 1998; Brechet et al., J Cell Biol 2008). Our mass-spectrometry experiments that relied, in part, on co-IP, used large amounts of protein and mass-spectrometry is now more sensitive than immunoblot. Finally, we do not know why the AnkR band in the PlxnA4 IP (Figure 2L) is slightly higher than for the AnkR or input IP. We agree that it is curious. It is possible that the smaller pool of PlxnA4-interacting AnkR is phosphorylated differently than the larger total pool of AnkR. Phosphorylation of membrane proteins can cause them to run at higher molecular weights (see for example Kv2.1 and Kv4.2 K^+^ channels – Misonou et al., Nat Neuro 2004, Figure 4).

4) Consistent quantification of the microscopy images would significantly strengthen conclusions. For example, the authors claim that somatodendritic NrCAM, but not that at the AIS, was decreased in Ank1-deficient mice (Figure 2J), but no quantification is provided (nor is another marker for the normally ankyrin-R-positive cells, see point 2). No description is provided for how it was determined if a cell was positive for markers like PV or WFA. It is unclear how were the images thresholded and counted.

Respectfully, we performed extensive quantification of images in Figures 3, Figure 3-Supplement 2, 6, and Figure 6-supplement 1 (formerly figure S6). We also now include a new Figure 3 – Supplement 1 showing the costaining between AnkR and WFA in control and AnkR cKO cortex. As the reviewer points out we did not quantify changes in perisomatic and AIS NrCAM. In revised Figure 2J, we report this comparison in three separate control and AnkR cKO mice. We include additional details in the methods for how thresholding was performed and cells counted.

5) The results shown here implicate ankyrin-R in the normal protein levels of Kv3.1b, with null mice showing ~50% decrease in channel levels. It is surprising that significant changes in potassium channel levels would not have an impact on resting membrane potential or input resistance.

Based on what is known about the Kv3.1 homozygous knock-out mice we are not surprised to see no change in resting membrane potential and input resistance. Porcello et al., (J Neurophysiol 2002) investigated the physiological phenotypes of Kv3.1 knock-out in Pv+ fast spiking cells of the RTN and also found no change in the resting membrane potential or input resistance. Furthermore, Lau et al., (J Neurosci 2000) showed that the resting membrane potential and input resistance are not altered in cortical fast-spiking PV interneurons of Kv3.2 KO. Thus, it’s not surprising that a 50% reduction in Kv3.1 did not result in these changes.

It would be useful to see ensemble potassium currents from these cells along with phase plane plots of the action potential.

Given the large number of potassium channels in these cells, it would not make sense to record ensemble potassium currents, as they are unlikely to be disrupted due to current from other types of potassium channels. Nevertheless, as requested we now include phase plane plots of the action potential with the data we have already collected. These are included in a revised Figure 5-supplement 1.

Similarly, it is surprising that loss of 50% of Kv3.1b does not have an impact on PV neuron fast-spiking, given the critical role of this channel in repolarization.

Complete Kv3.1 knock out in either fast spiking RTN neurons or fast spiking neurons of the auditory brainstem has very little or no effect on the firing frequency of these fast-spiking cells, but rather results in a similar phenotype to what we report here, with a significant increase in the failure rate at high currents (Porcello et al., J Neurophysiol 2002; Macica et al., J Neurosci 2003).

It is unclear whether there is compensation from Kv3.1a.

Nevertheless, we obtained pan Kv3.1 antibodies which recognize BOTH Kv3.1a and Kv3.1b. We found significant loss of Kv3.1 immunoreactivity in AnkR cKO mouse brains. These data are now shown in new Figure 6 -Supplement 1D and 1F. In addition, we performed immunostaining for Kv3.2 to check for compensation. As for Kv3.1b, we observed a similar reduction in Kv3.2 immunoreactivity. These results are now included in new Figure 6 -Supplement 1E and 1G.

Reviewer #2:The paper by Stevens et al. examines the role of ankyrin R in parvalbumin-expressing, fast-spiking GABAergic interneurons. To address this question, the authors use immunocytochemistry, proteomics, and electrophysiology. The main findings are:- Deletion of ankyrin R disrupts perineuronal nets, decreases anxiety-like behaviors, and changes the intrinsic excitability of PV+ interneurons.- Deletion of ankyrin R leads to a significant reduction of Kv3.1 potassium channel density.- Interaction between ankyrin R and Kv3.1 occurs via a novel ankyrin binding motif.Based on these results, the authors conclude that ankyrin R regulates function of PV+ interneurons by organizing cell adhesion molecules, perineuronal nets, and ion channels. Overall, this is a potentially interesting paper. Although the role of perineuronal nets and Kv3 channels in the excitability and fast spiking of PV + interneurons is well established, in the finding that ankyrin R regulates these properties seems novel.1. The conclusion that input synapses of PV+ interneurons are unaffected by ankyrin R is on weak grounds, and is partially contradictory with previous literature. Ideally, the authors should perform paired recording experiments to address this issue.

The reviewer is correct and we cannot claim that input synapses are unaffected. We removed any claim about altering synaptic inputs.

2. The conclusion that the perineuronal net is more fragmented after ankyrin R deletion is unconvincing. High-resolution imaging followed by three dimensional reconstruction seems necessary. Ideally, super-resolution imaging should be performed.

To overcome this issue, we now include a revised Figure 3 showing measurement of the ‘thickness’ of the net surrounding the neuron at both 1 and 12 months of age. As shown in the results, the AnkR cKO nets are significantly less compact than in control neurons. To better demonstrate the extensive holes found in the perisomatic net we include a revised Figure 3 – Supplement 2 (formerly Supplemental Figure 3) showing WFA z-projections at 3 different planes of representative neurons at 1 and 12 months of age. These images show the presence of large holes and a more diffuse net in the AnkR cKO neurons. We agree that super-resolution imaging and reconstruction like that done in Sigal et al., (PNAS 2019) would be ideal. In that study, the authors used super-resolution microscopy to generate a Behrmann equal-area cylindrical surface projection to measure hole size in perineuronal nets. We would love to do that analysis. Unfortunately, it is at the moment beyond our technical capabilities. We hope the additional images and results included in this revision satisfy this reviewer’s concerns.

3. The authors describe several links between ankyrin R and Kv3.1. However, other channels may be affected as well. First, ankyrin R may affect sodium channel density in interneurons. This important point is unaddressed in the present paper.

We agree with this point and did not address Na^+^ channel densities. We previously reported that AnkR can bind to Na^+^ channels, although with much lower affinity than for AnkG (Ho et al., Nat Neuro 2014). Since AnkG’s distribution is not affected and AnkR is not present at the axon initial segment (AIS) (supplemental Figure 1B and Liu et al., *eLife* 2020), AIS Na^+^ channels remain unaffected. Nevertheless, the loss of AnkR from perisomatic regions could result in the reduction of perisomatic Na^+^ channel density. Unfortunately, it is technically extremely difficult to measure Na^+^ channel density by immunohistochemical methods since Na^+^ channels are present in perisomatic domains at densities too low to be detected by immunofluorescence. I am unaware of any measurements of perisomatic Na^+^ channel densities measured using antibodies except for that done using freeze fracture immunolabeling of cortical pyramidal neurons (Lorincz and Nusser, Science 2010). Although we believe the analysis of Na^+^ channel densities in perisomatic regions of interneurons is beyond the scope of this paper, we performed immunostaining of interneurons using a Pan Nav antibody and find no difference between WT and AnkR cKO interneurons at both AIS and in perisomatic regions. We include a representative image in Author response image 1 with quantification for the reviewers (please keep in mind the caveats and difficulty of measuring perisomatic Na^+^ channel densities since they are not above background). However, these results do not mean there are no changes, only that with our current tools we cannot detect these changes. We modified the Discussion section to include the possibility that there may be changes in the density of perisomatic Na^+^ channels.

**Author response image 1. sa2fig1:** 

Second, Kv 3.2 may play a role in regulation of excitability, independently of the reported effects. Kv3.2 Is functionally redundant with Kv 3.1, and is coexpressed with Kv3.1 in several types of GABAergic interneurons. Clearly, the expression of these two major players would need to be addressed.

The reviewer is correct that Kv3.2 is also found in AnkR+ interneurons. According to their suggestion, we performed immunostaining of Kv3.2 in control and AnkR cKO mouse brain. As shown in Figure 6 – Supplement 1E and 1G, we observed a similar reduction in immunoreactivity like that seen for Kv3.1b. This is not surprising since Kv3.1 and Kv3.2 K^+^ channel subunits form heteromers when expressed in the same cells. Our results (Figure 6I) strongly suggest that Kv3.2 is localized due to its association with AnkR through Kv3.1b.

4. The authors claim that 70% of ankyrin R-positive neurons are immunopositive for parvalbumin. The identity of the remaining 30% is unclear. From my point of view, it would be important to fill this gap in the authors' conclusion. Immunolabeling for somatostatin could provide a quick answer.

We agree with this reviewer that it is an important point. We performed a detailed analysis of the AnkR+ cells that are Pv+/SST+, Pv+/SST-, Pv-/SST+, and Pv-/SST-. We performed this analysis for both cortex and hippocampus. These new results are shown in Figures 1C and 1D, with triple immunostaining of cortex shown in a new Figure 1 – Supplement 1A. While ~85% of AnkR+ cells are Pv+ or SST+, ~15% of AnkR+ cells remain unidentified. We know they are forebrain interneurons of some type since AnkR immunoreactivity is lost in DLX5/6-cre mice.

5. The subcellular distribution of ankyrin R in PV+ GABAergic interneurons is insufficiently analyzed. If the authors want to make the point that ankyrin is located "somatodendritically", they would need to quantify the expression in dendrites.

We agree with this reviewer that our use of the term somatodendritic is incorrect. We now use the term ‘perisomatic’ which we agree is a much better term.

6. Functional conclusions stand and fall with the analysis of the current-clamp data. Conclusions may be corroborated by voltage-clamp analysis of Kv3 currents, preferentially under optimal voltage-clamp conditions in nucleated patches. Such experiments may reveal whether current density, channel gating, or both are changed.

We agree that this information would be quite interesting. Since our immunostaining and biochemical results show significant reductions in Kv3.1b and Kv3.2, we think the simplest explanation is that there is a reduction in channel/current density. However, we cannot rule out a change in channel gating. In fact, one single study examined the effect of ankyrin-binding on Na^+^ channel gating (Shirahata et al., J Neurophysiol 2006). That study showed AnkG, but not AnkB, dramatically affected persistent Na^+^ current. Thus, different ankyrins may have different effects on channel gating and the effect will almost certainly also be channel specific. Respectfully, we believe this is beyond the scope of this paper.

7. There are some concerns about statistics. Line 815 states: "Data distributions were assumed to be normal" , but clearly the validity of this assumption has to be tested. Furthermore, in line 1198, the authors state that an ANOVA or a (nonparametric) Kruskal-Wallis test was used. These discrepancies should be resolved. Finally, it would need to be stated more clearly if and how correction for multiple comparisons was done in the pairwise tests.

Please also see response to reviewer #1, above. Briefly, data distributions are now not assumed to be normal and this is now stated in the methods. In the case of electrophysiological studies normality was explicitly tested. Correction for multiple comparisons is fully described and reported for each data set in the extended data.